# X-Diffusion: Generating Detailed 3D MRI Volumes From a Single Image Using Cross-Sectional Diffusion Models

## Abstract

Magnetic Resonance Imaging (MRI) is a crucial diagnostic tool, but high-resolution scans are often slow and expensive due to extensive data acquisition requirements. Traditional MRI reconstruction methods aim to expedite this process by filling in missing frequency components in the K-space, performing *3D-to-3D* reconstructions that demand full 3D scans. In contrast, we introduce *X-Diffusion*, a novel cross-sectional diffusion model that reconstructs detailed 3D MRI volumes from extremely sparse spatial-domain inputs—achieving *2D-to-3D* reconstruction from as little as a single 2D MRI slice or few slices. A key aspect of X-Diffusion is that it models MRI data as holistic 3D volumes during the cross-sectional training and inference, unlike previous learning approaches that treat MRI scans as collections of 2D slices in standard planes (coronal, axial, sagittal). We evaluated X-Diffusion on brain tumor MRIs from the BRATS dataset and full-body MRIs from the UK Biobank dataset. Our results demonstrate that X-Diffusion not only surpasses state-of-the-art methods in quantitative accuracy (PSNR) on unseen data but also preserves critical anatomical features such as tumor profiles, spine curvature, and brain volume. Remarkably, the model generalizes beyond the training domain, successfully reconstructing knee MRIs despite being trained exclusively on brain data. Medical expert evaluations further confirm the clinical relevance and fidelity of the generated images. To promote reproducibility and trust in our findings, we will publicly release the accompanying code upon publication. To our knowledge, X-Diffusion is the first method capable of producing detailed 3D MRIs from highly limited 2D input data, potentially accelerating MRI acquisition and reducing associated costs.

## 1 Introduction

Medical imaging stands as a cornerstone in modern healthcare, with innovations playing a critical role in disease diagnosis and treatment planning. Traditional MRI scans, though detailed, are often time-consuming and come with significant economic implications (Bell, 1996). The urgency to tackle these impediments has propelled research endeavors, but the quest for a cost-efficient, rapid, and precise alternative persists (Wald et al., 2020; Arnold et al., 2023; Sarracanie et al., 2015). A rapid and affordable MRI process would catalyze early disease detection, potentially saving countless lives. Moreover, by reducing barriers to access, we would ensure a more holistic healthcare approach, promptly addressing diseases before they escalate.

Traditionally, inverse 2D or 3D Fast Fourier Transform (FFT) (Brigham & Morrow, 1967) on k-space data with full Cartesian sampling is used to reconstruct MR images from raw data, sometimes with the help of machine learning models (Fessler, 2010; Tran-Gia et al., 2013; Roeloffs et al., 2016; Ben-Eliezer et al., 2016; Tan et al., 2017; Wang et al., 2018). Recent years have seen a pivot towards machine learning-based frameworks such as Generative Adversarial Networks (GANs) and diffusion-based models, harnessing the power of deep neural networks to enhance MRI reconstruction (Quan et al., 2017; Jiang et al., 2021; Chung et al., 2022). However, a pervasive challenge remains: the synthesis of high-resolution MRIs from extremely limited observations (or even a single 2D image). Previous works either target compressive sensing to increase the frequency resolution of the MRI (Chung et al., 2023; Quan et al., 2017; Jiang et al., 2021; Chung & Ye, 2022) or aim to

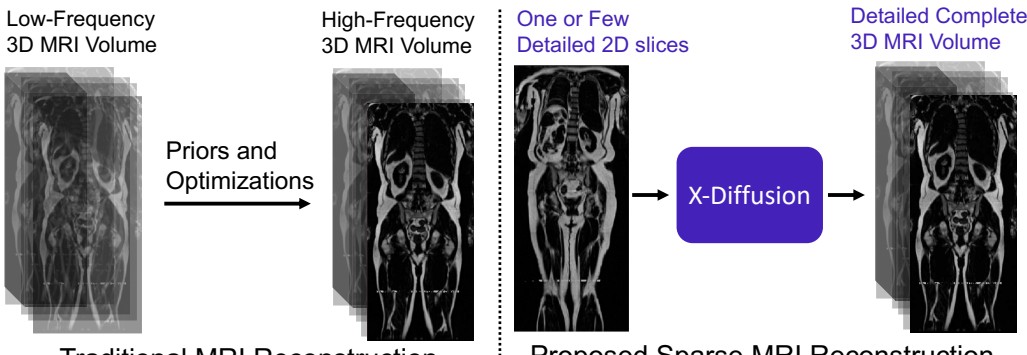

Figure 1: **X-Diffusion for Sparse MRI Reconstruction**. (*Right*) We present X-Diffusion, a method that can generate detailed and dense MRI volumes from a single MRI slice or a few slices. X-Diffusion is the first method in medical imaging to generate detailed 3D MRIs from extremely sparse inputs, preserving key anatomical properties. (*Left*) MRI reconstruction traditionally involves retrieving high-frequency images from low-frequency full 3D MRI volumes (in the K-space).

increase the slice density when a sufficient number of slices is available (more than 30) (Lee et al., 2023). These existing gaps in the MRI reconstruction landscape underscore the significance of our approach in reconstructing MRIs from an extremely small number of observations.

Motivated by this, we propose *X-Diffusion*, a novel architecture that learns on 3D volumetric data by utilizing view-dependent cross-sections. This approach allows for full MRI generation with high accuracy from a single MRI slice or multiple slices (see Figure 1). Unlike previous methods that treat MRI data as collections of 2D slices in standard planes (coronal, axial, sagittal) or rely heavily on frequency-domain data, X-Diffusion operates directly in the spatial domain and models MRI samples as complete 3D volumes during both training and inference. To our knowledge, X-Diffusion is the first method capable of producing detailed 3D MRIs from highly limited 2D input data, potentially accelerating MRI acquisition and reducing associated costs. It is important to note that the generated MRIs are not clinical replacements for true MRIs yet, but could provide a quick, affordable, and informative "pseudo-MRI" before conducting a full MRI examination.

**Contributions:** **(i)** We introduce X-Diffusion, a cross-sectional diffusion model that generates MRI volumes conditioned on a single input MRI slice or multiple slices. The proposed X-Diffusion achieves state-of-the-art results on MRI reconstruction and super-resolution compared to recent methods on BRATS, a large public dataset of annotated MRIs for brain tumors, and full-body MRIs from the UK Biobank dataset. **(ii)** We validate the generated MRIs on a wide range of tasks to ensure that they retain important features of the original MRIs (*e.g.*, tumor profiles and spine curvature) without using this meta-information in the generation process. **(iii)** We showcase the generalization of trained X-Diffusion beyond the training domain (*e.g.*, on knee MRIs not seen in training). **(iv)** We evaluated the generated brain and knee MRIs with medical experts (a surgeon and an oncologist) who anonymously could not distinguish the real from the generated MRIs in controlled experiments which provides a proof of concept for the potential clinical usefulness of the generated MRIs.

## 2 RELATED WORK

**Single-View 3D Reconstruction.** Recent efforts on predicting 3D from 2D RGB images are starred with the seminal work of DreamFusion (Poole et al., 2022), which distilled a ready-made diffusion mechanism (Saharia et al., 2022) into NeRF (Mildenhall et al., 2020; Barron et al., 2022). This methodology ignited a myriad of new techniques, both in converting text to 3D ((Lin et al., 2023; Chen et al., 2023)) and transitioning visuals to 3D forms ((Melas-Kyriazi et al., 2023; Liu et al., 2023; Tang et al., 2023; Qian et al., 2023)). These frameworks were considerably improved by Zero-123 (Liu et al., 2023), explicitly conditioning on camera-views while finetuning Stable Diffusion on the large 3D CAD dataset Objaverse (Deitke et al., 2023). While Zero-123 learns to generate surface renderings of a target view given a single image, X-Diffusion learns to generate a cross-sectional slice, conditioned on the angle and depth index of the slice, allowing for dense 3D volume generation and targeting MRI medical imaging.

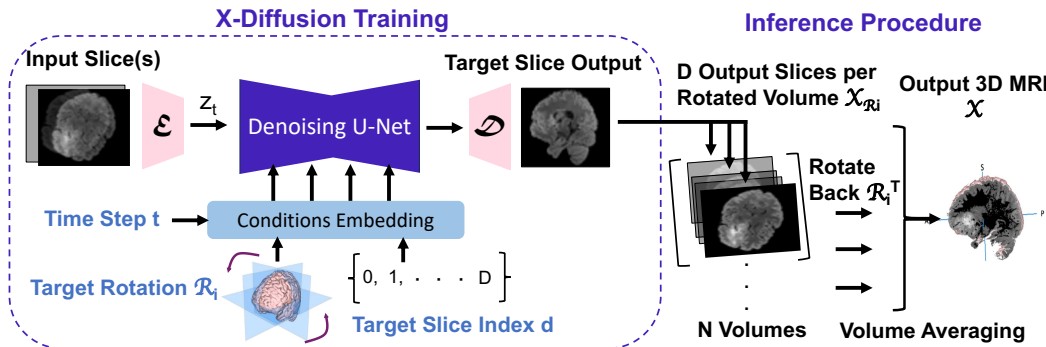

Figure 2: **X-Diffusion Pipeline**. A single or multi-slice input is fed into the Latent Diffusion U-Net conditioned on the target slice index $d$ and target rotation from 360° slicing. The 3D volume is reconstructed by vertical stacking of the slices from a fixed axis of rotation. The final volume $\mathcal{X}$ is obtained after averaging the $N$ realigned view-dependant volumes $R_i^\mathsf{T} \mathcal{X}_{R_i}$ from a set of predefined target rotations $R_i$.

**Full-Body MRI Analysis.** Most methods on automatic MRI analysis focused on developing methods for local segmentation of organs or tumours (Chen et al., 2019; Doran et al., 2017; Windsor & Jamaludin, 2020; Ranjbarzadeh et al., 2021). Relatively few studies looked at whole-body scans. Most of them were developed to detect and segment the spine in tasks such as scoliosis detection(Jamaludin et al., 2017; 2020; Windsor et al., 2020; 2021; Bourigault et al., 2022). In their 2020 article, Tunariu *et al.* discuss advancements in whole-body magnetic resonance imaging (MRI) and its applications in clinical practice (Tunariu et al., 2020). Similarly, Küstner *et al.* present a deep learning approach for the automatic segmentation of adipose tissue in whole-body MRI scans, facilitating large-scale epidemiological studies (Küstner et al., 2020). We leverage the MRI analysis techniques for validating the viability of the generated MRIs for tumor, spine, and other discriminative features of interest.

**MRI Reconstruction.** With the recent rise of foundation models in computer vision (Rombach et al., 2022a; Caron et al., 2021; OpenAI, 2023), several attempts have shown promise in steering these models for the medical imaging domain (Ma & Wang, 2023; Nguyen et al., 2023). However, this is mainly limited to discriminative tasks such as segmentation, classification and detection. For Medical imaging inverse problem tasks, mostly classical methods were employed for incensing the resolution of the reconstruction (Ronneberger et al., 2015; Schlemper et al., 2017; Shi et al., 2015; Wang et al., 2014), or adopt diffusion models without great leverage of image pretraining (Chung & Ye, 2022; Chung et al., 2023; Lee et al., 2023; Song et al., 2021). The LRTV method combines low-rank and total variation regularizations to enhance the resolution of MRI images Shi et al. (2015). This approach effectively preserves image details while reducing noise, leading to improved image quality. A similar work presents super-resolution MRI based sparse reconstruction framework, by proposing a simultaneous two-dictionary training method for sparse reconstruction Wang et al. (2014). ScoreMRI and TPDM (Chung & Ye, 2022; Lee et al., 2023) make use of diffusion probabilistic model (DPM) performing conditional sampling-based inverse problem. TPDM (Lee et al., 2023) proposed to overcome the limitation of ScoreMRI being an image-to-image model and leveraged the 3D prior distribution of the data using a product of two 2D diffusion models. Although this approach enables 3D generation, it only samples from two fixed canonical planes from the 3D MRI and does not work for sparse input. On the other hand, X-Diffusion leverages the full 3D volume by sampling the brain in all directions and leverages the Stable Diffusion huge image pretraining for 3D MRI volumes from a single MRI slice.

## 3 METHODOLOGY

Our approach is delineated into three primary aspects: conditioning of the diffusion model, denoising cross-sectional slices, and slice stacking of view-conditioned volumes to generate the final MRI output (as shown in Figure 2).

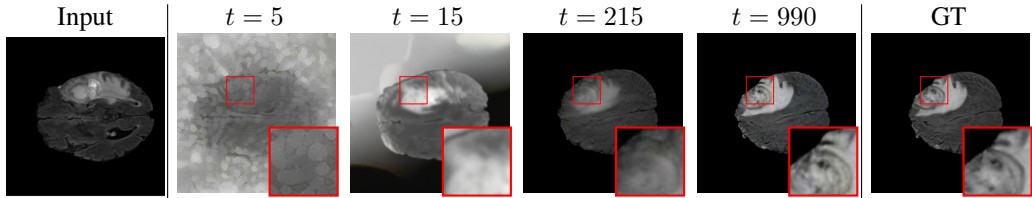

Figure 3: **Test Time Brain Generation at Different Sampling Steps**. For the input slice index of 107 (*left*), we show the ground-truth slice 90 (*right*) and the corresponding brain slice generated at different sampling steps $t$ in the denoising diffusion process of X-Diffusion trained on BRATS.

## 3.1 DIFFUSION MODELS PRELIMINARIES

In previous works on view-conditional diffusion (Kawar et al., 2022; Rombach et al., 2022a; Liu et al., 2023), the diffusion model $\epsilon_\theta$ is trained based on the objective:

$$\min_\theta \mathbb{E}_{z \sim \mathcal{E}(x), t, \epsilon \sim \mathcal{N}(0,1)} ||\epsilon - \epsilon_\theta(z_t, t, c(x, R, \tau))||_2^2 \tag{1}$$

In Equation 1, $\theta$ denotes the model parameters that are being optimized. The latent variable $z$ is sampled from a distribution $\mathcal{E}(x)$, where $x$ indicates the input data and $\mathcal{E}$ and $\mathbb{D}$ are the pretrained frozen encoder and decoder of LDM AE respectively Rombach et al. (2022a). $t \sim [1, 2, ..., T]$ specifies a particular time step during the diffusion process with maximum $T$ steps. The term $\epsilon$ is a noise variable sampled from a standard normal distribution, $\mathcal{N}(0, 1)$. The function $\epsilon_\theta$ is representative of the model's prediction for a given $z_t, t$, and transformation $c(x, R, \tau)$, where $R$ and $\tau$ are rotation and translation parameters, respectively.

Proceeding, the gradient of the Score Jacobian Chaining (SJC) loss, which approximates the score towards the non-noisy input as described in (Liu et al., 2023; Rombach et al., 2022a), is given by: $\nabla \mathcal{L}_{SJC} = \nabla_{I_\pi} \log p_{\sqrt{2}\epsilon}(x_\pi)$. The term $\nabla_{I_\pi}$ specifies the gradient with respect to the image $I_\pi$. The expression $p_{\sqrt{2}\epsilon}(x_\pi)$ denotes the probability distribution of the transformed image $x_\pi$ under noise level $\sqrt{2}\epsilon$. In our setup, $\tau$ is replaced with the index $d$ of the slice of the MRI volume, and $R$ is the rotation applied to the MRI volume for the cross-sectional processing.

## 3.2 X-DIFFUSION FOR CROSS-SECTIONAL MRI SYNTHESIS

Upon acquiring the MRI slice $x \in \mathbb{R}^{H \times W}$, we seek to synthesize the entire MRI volume $\mathcal{X} \in \mathbb{R}^{H \times W \times D}$. For this, we employ X-Diffusion $\epsilon_\theta$, a cross-sectional diffusion model. The fundamental idea stems from the analogy that a 3D volume can be built crosswise by stacking slices from a certain direction, just like a loaf of bread. The full target volume $\bar{\mathcal{X}}$ can be reconstructed from limited slices by generating target slices indexed by their depth $d \in [1, 2, ..., D]$ in the MRI volume conditioned on a certain direction $R$ where the volume is oriented. This simplifies the learning of cross-sections since the rotated MRI volume $R\mathcal{X}$ will have the same size $H \times W \times D$ as the original volume where zero padding is used. For simplicity of the processing of the data, we use the same dimensions for all directions ($H = W = D$). This allows varying the depth after rotating the ground truth MRI $\bar{\mathcal{X}}$ volume by simply indexing by the depth index $d$, and hence the slice that is used for training will be $\bar{x}_d = (R\mathcal{X})_{d,:,:}$. The full objective of training X-Diffusion is as follows.

$$\min_\theta \mathbb{E}_{z,t,\epsilon,d,R} ||\epsilon - \epsilon_\theta(z_t, t, c(x, d, R))||_2^2$$
$$\text{s.t.} \quad z \sim \mathcal{E}(\bar{x}_d), \ \ t \sim [1, 2, ..., T] \tag{2}$$
$$\epsilon \sim \mathcal{N}(0, 1), \ \ d \sim [1, 2, ..., D], \ \ R \sim SO(3)$$

The X-Diffusion model is trained with cross-sections from all different directions $R$ and all different depths $d$, which allows it to generate the target from any arbitrary rotation and depth (see Figure 3). At inference, unrolled X-Diffusion is applied $D$ times with $d \in [1, 2, .., D]$ from an arbitrary orientation $R_i$, and decoded with decoder $\mathbb{D}$ to obtain the *view-conditional volume* $\mathcal{X}_{R_i}$. This volume is then rotated back by $R_i^\mathsf{T} \mathcal{X}_{R_i}$ to the Canonical orientation to produce the final output MRI $\mathcal{X}$.

$$\mathcal{X}_{R_i} = \begin{bmatrix} \mathbb{D}\Big(\epsilon_\theta(z_t, t, c(x, 1, R_i))\Big) \\ \vdots \\ \mathbb{D}\Big(\epsilon_\theta(z_t, t, c(x, D, R_i))\Big) \end{bmatrix}, \ \ t = 1, 2, ..., T \ , \ \ \ \mathcal{X} = R_i^\mathsf{T} \mathcal{X}_{R_i}. \tag{3}$$

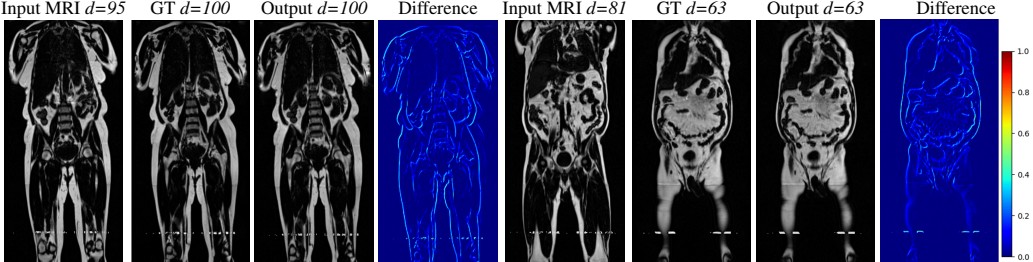

Figure 4: **Qualitative Results of full body 3D MRI Generation with X-Diffusion.** We show a single MRI slice example, two corresponding ground-truth MRI slices (index 68 and 100), the corresponding generated MRI slice, and a difference map to qualitatively measure the error between generated and ground-truth MRI.

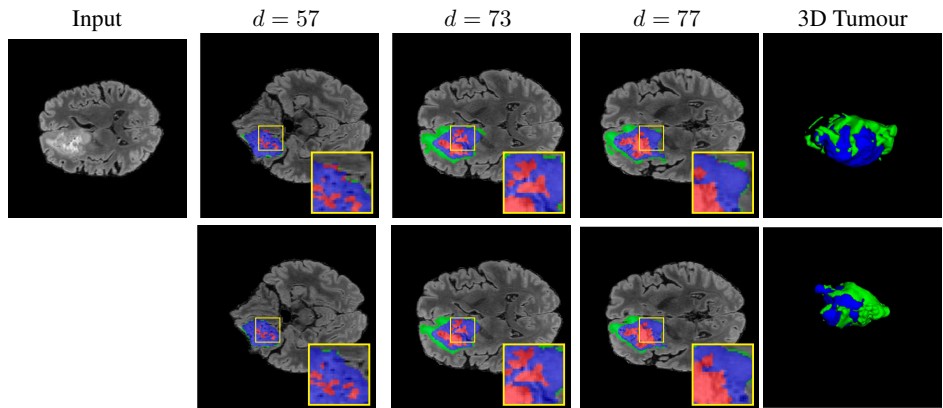

Figure 5: **Visualisations of 3D Brain Generation**. For the input slice (slice index 76), we show examples of slices from generated 3D brain MRI volumes with varying slice index (*top*) and its ground-truth brain slices (*bottom*). We show the tumour profile segmentation map in all output and ground truth slices to highlight the differences and show the 3D tumor in the generated MRI and ground truth MRI in the most right column. Red is used for non-enhancing and necrotic tumor core, green for the peritumoral edema, and blue for the enhancing tumor core.

**Multi-Slice Input.** While the pipeline described above is effective, it relies on heavy diffusion operations for each slice input and output. Adding more slices by simply inflating the network (Blattmann et al., 2023) will create computational and memory difficulties. Therefore, to efficiently allow X-Diffusion's pipeline to accept $K$ slices as input while maintaining the same original weights structure of Stable Diffusion (Rombach et al., 2022a), we perform a cumulative sum operation on the dot product of consecutive slices to reduce to a single slice input. The reduction operation of the $K > 1$ input slices $(x_1, x_2, ..., x_K)$ is similar to what is followed in TPDM (Lee et al., 2023) in the conditioning volume, and it can be described as follows. $x = \frac{1}{K-1} \sum_{j=1}^{K-1} x_j \cdot x_{j+1}$. These multi-slice inputs can be from the same plane (our experiments' focus) or orthogonal planes. We compare in Section 6.2 the different strategies to process the slices with different configurations.

**Multi-View MRI Volume Generation.** One advantage of our cross-sectional diffusion is that it can learn and generate the volume $\mathcal{X}_{R_i}$ from any arbitrary view direction $R_i$ (as in Equation 3). In training, this allows X-Diffusion to train on MRIs from all types of cross-sections, unlike the typically followed common 3 planes (coronal, sagittal, and axial) (Chung & Ye, 2022; Lee et al., 2023; Fessler, 2010), which allows the model to generalize better. At inference, we leverage this power to generate $N$ volumes from $N$ different views predefined as equally distributed views around the $360°$ around the azimuth horizontal rotations $R_i \in \{R_{\text{azim}}(\frac{i \times 360°}{N})\}_{i=1}^{N}$, where $R_{\text{azim}}(r)$ is the rotation matrix defined by rotating by $r$ degrees around the vertical axis $(0,1,0)$. The final MRI volume output $\mathcal{X}$ is then obtained by averaging the view-conditional volumes ($\mathcal{X}_{R_i}$ from Eq (equation 3)) at inference after rotating back to the canonical orientation of the output as follows. $\mathcal{X} = \frac{1}{N} \sum_{i=1}^{N} R_i^{\mathsf{T}} \mathcal{X}_{R_i}$ This multi-view aggregation is inspired by how multi-view discriminative methods learn a global representation by aggregating multiple views features(Su et al., 2015; Hamdi et al., 2023). We show in Section 6.1 the utility of the volume averaging compared to a single volume.

| Models | Test 3D PSNR ↑ | | | | | | | | | | | |
| | 1 slice | | 2 slices | | 3 slices | | 5 slices | | 10 slices | | 31 slices | |
| | **BR** | **UK** | **BR** | **UK** | **BR** | **UK** | **BR** | **UK** | **BR** | **UK** | **BR** | **UK** |
|---|---|---|---|---|---|---|---|---|---|---|---|---|
| ScoreMRI | 9.37 | 8.54 | 10.25 | 9.16 | 10.68 | 10.42 | 12.37 | 11.88 | 14.31 | 13.24 | 29.24 | 19.01 |
| TPDM | 10.48 | 9.29 | 10.86 | 9.99 | 11.33 | 11.09 | 14.13 | 12.62 | 16.65 | 15.88 | 31.48 | 21.70 |
| X-Diffusion (ours) | **23.10** | **22.42** | **25.20** | **23.04** | **29.43** | **25.26** | **31.25** | **26.85** | **33.27** | **27.44** | **35.48** | **29.01** |

Table 1: **Model Performance on Test Brain Data and Whole-Body MRIs**. We compare the MRI reconstruction for baselines ScoreMRI (Chung & Ye, 2022), TPDM (Lee et al., 2023), and our X-Diffusion model for varying input slice numbers in training and inference. We report the mean 3D test PSNR on BRATS (**BR**) brain dataset and the UK Biobank body dataset (**UK**). The results showcase huge improvement over the baselines, especially on the small number of input slices (particularly at 1). For reference, the parameter count and inference time for processing a single 3D MRI on a single NVIDIA A6000 GPU with 48GB of RAM are as follows: ScoreMRI (860M, 139.1s), TPDM (1720M, 149.5s), and X-Diffusion (990M, 141.5s).

## 4 EXPERIMENTS

### 4.1 DATASETS

We conducted our experiments on two primary datasets for evaluations (BRATS & UK BioBank) and on two secondary datasets for out-of-domain generalization (IXI & fast knee MRI). **BRATS** is the largest public dataset of brain tumours consisting of 5,880 MRI scans from 1,470 brain diffuse glioma patients, and corresponding annotations of tumours (Baid et al., 2021; Menze et al., 2015a; Bakas et al., 2017). All scans were skull-stripped and resampled to 1 mm isotropic resolution. All images have a resolution of $240 \times 240 \times 155$, and we use the flair T2 sequence. Tumours are annotated for 3 classes: Whole Tumour (WT), Tumour Core (TC), and Enhanced Tumour Core (ET). **UK Biobank** is a more comprehensive dataset of 48,384 full-body MRIs from more than 500,000 volunteers(Sudlow et al., 2015), capturing diverse physiological attributes across a broad demographic spectrum. **IXI** is a dataset of T1-weighted 1.5 Tesla brain MRI images of 582 healthy subjects, freely available online (IXI). **Knee fastMRI** is a public dataset of raw k-space data from NYU Langone(Knoll et al., 2020; Zbontar et al., 2019). We use the test set provided (n=109) of fastMRI single coil, dimensions 640x372x30. These are center-cropped to 320x320x30.

### 4.2 EVALUATION METRICS

We use the standard 3D **PSNR** (Lee et al., 2023) and 2D **SSIM** (Wang et al., 2004) metrics to evaluate 3D MRI reconstruction and the following metrics for the validation experiments. **Dice Score** is used to evaluate the performance of our model at segmenting the brain tumours (Menze et al., 2015b). Dice Score $= \frac{2|Y \cap \hat{Y}|}{D(|Y| + |\hat{Y}|)}$, where $Y$ is the prediction, $\hat{Y}$ is the ground-truth label and $D$ the total number of slices. **Brain Volume.** We measure brain volume in $mm^3$ by counting the non-zero voxels in the volume multiplied by the voxel spacing (Dikici et al., 2019). **Spine Curvature.** Let $\gamma(t) = (x(t), y(t))$ be the equation of a twice differentiable plane curve parametrized by $t \in [1, 209]$ normalized height-wise by 209 for curvature analysis. We measure the spine curvature $\kappa$ similar to (Bourigault et al., 2023): $\kappa = (y'' x' - x'' y')/(x'^2 + y'^2)^{\frac{3}{2}}$.

### 4.3 BASELINES

We compare X-Diffusion's performance against state-of-the-art MRI generation techniques, namely ScoreMRI (Chung & Ye, 2022) and Two-Perpendicular-Diffusion-Models TPDM (Lee et al., 2023) using NCSNPP model (Song et al., 2020). For the multiple slice input (nx256x256) in X-Diffusion, we aggregated the multiple inputs to form a single batch (1x256x256). For comparison with ScoreMRI, being an image-to-image model, we uniformly sampled $n$ slices along the z-axis. As for TPDM, we conditioned on $n$ slices from the full volume after the fusion of the two diffusion models.

### 4.4 IMPLEMENTATION DETAILS

To facilitate using the pretrained weights of Zero-123 (Liu et al., 2023) (based on Stable Diffusion (Rombach et al., 2022a)), we use the same channel size in the input 3, repeating the grayscale images. For the size of the MRI volumes, we used $H = W = D = 155$, as originally the sizes in

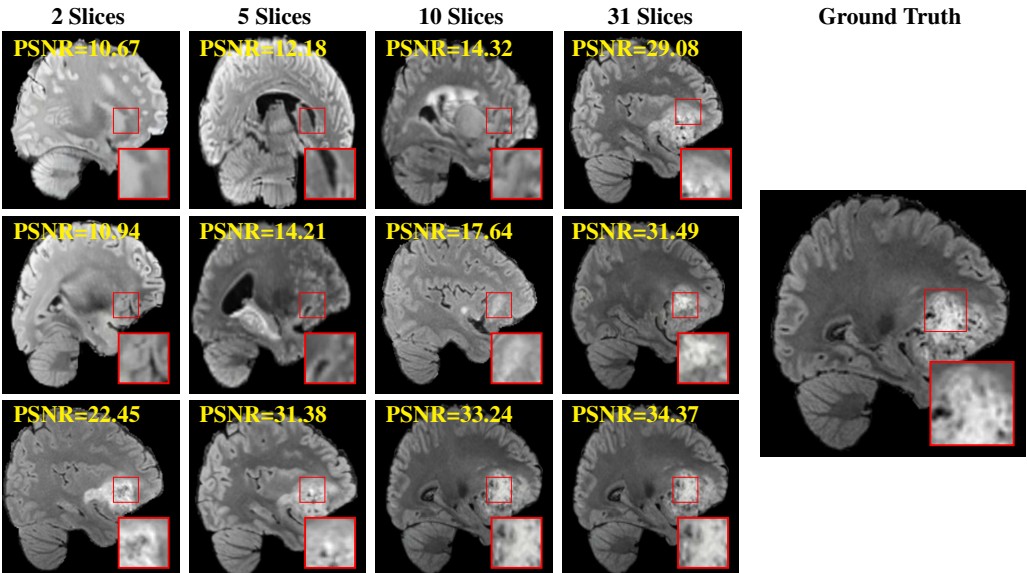

Figure 6: **Visual Comparison of MRI Brain Reconstruction.** We benchmark different methods of reconstructed 3D brains on test set with multi-slice inputs. We show a generated slice from 3D brain generated from ScoreMRI (Chung & Ye, 2022) (*top*), TPDM(Lee et al., 2023) (*middle*), and X-Diffusion (*bottom*) conditioned on a varying number of input slices. The red zoomed crop is placed in the exact location in all images to highlight the differences.

the dataset were 155 slices. For model training, we use a base learning rate of $1.0e^{-06}$. Batch size is set to 32. In the diffusion sampling, we used $T = 1000$ time steps and an ETA of 1.0. More details about the datasets, metrics, and setup are provided in *Appendix*.

## 5 RESULTS

### 5.1 MAIN RECONSTRUCTION RESULTS

Our results unequivocally highlight the superior performance of X-Diffusion in terms of both qualitative and quantitative metrics. Representative MRI volumes generated by our pipeline, when juxtaposed with ground-truth images, showcased remarkable similarity, with even intricate physiological features like tumor information, spine curvature, and fat distribution being accurately captured.

Notably, X-Diffusion achieves *sota* $PSNR > 30$ dB for a few input slices while baselines require more than 60 input slices to achieve similar performance (Figure 7). The margin is more than 12 dB PSNR for the 1-slice input in both the BRATS and the UK Biobank benchmarks (see Table 1 and Figure 6). For reference, two randomly sampled MRIs from the UK Biobank would have a PSNR of 15.95 dB $\pm$ 0.36 (on 4800 randomly sampled examples). The slices from 3D reconstructed volumes at varying depths and axis of rotation, visually match the ground truths for both brain and whole-body scans (see Figures 5 and Figure 4). We also plot the error map (Figure 4) of such X-Diffusion generations to highlight the differences with the ground truth MRIs.

### 5.2 MRI VALIDATION RESULTS

**Brain Volumes Preservation.** The generated MRIs by our X-Diffusion retain almost the exact same average brain volume $1.28e^6$ $mm^3$ *vs* $1.31e^6$ $mm^3$ of the real MRIs.

**Tumour Information Preservation.** For the brain tumor segmentation, we use a Swin UNETR model(Hatamizadeh et al., 2022; Tang et al., 2021), trained with random rotation, and intensity as data augmentation. In Figure 5, we highlight the tumor profiles of the generated MRIs compared to the ground truth tumour profile. In the test set with human ground-truth annotations ($n = 333$), the real MRI Dice score is 85.15 while the generated MRIs from a single slice have a dice score of 83.09. This shows how the generated MRIs indeed preserve the tumor information and can act as

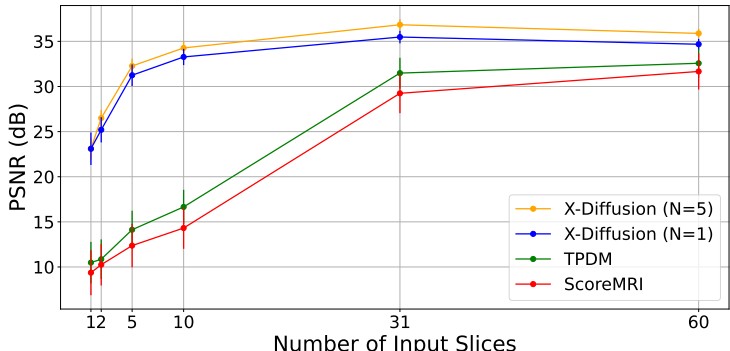

Figure 7: **Effect of the Number of Input Slices**. We plot the test PSNR *vs.* the number of input slices for X-Diffusion and our baselines i.e. TPDM (Lee et al., 2023) and ScoreMRI (Chung & Ye, 2022) on the brain MRI dataset. $N$ is the number of averaged view-dependent volumes. We show the standard deviation of each run to account for potential randomness.

an affordable and informative pseudo-MRI, before conducting an actual costly MRI examination in hospitals. More detailed results are provided in *Appendix*.

**Preservation of Spine Curvature.** For the spine segmentation on UK Biobank, we use a UNet++ model (Zhou et al., 2018) with Dice Loss and use the curvature prediction of the spine followed in (Bourigault et al., 2022)). We measure the Pearson correlation factor (Bourigault et al., 2022) of spine curvature measured on the generated MRIs where the input is a single MRI coronal slice, or a single sagittal slice against the curvature of reference real MRIs of the same samples. The correlation coefficients are 0.89 for the coronal MRIs and 0.88 for the sagittal MRIs on the test set of 308 human-annotated angles.

## 5.3 OUT-OF-DOMAIN GENERALISATION

One way to test the generalization capability of the trained X-Diffusion is to test it on a completely different domain from an MRI dataset not seen during training. We report the single-slice results on the test set of $n = 109$ knees from NYU fastMRI Knoll et al. (2020); Zbontar et al. (2019), using the X-Diffusion trained on the BRATS brain MRIs. The test PSNR result is 34.17 and an example is shown in Figure 8. More detailed results can be found in Table II of the *Appendix*. It shows how successfully X-Diffusion can generate knee MRIs (out-of-domain) despite being trained on brains.

## 5.4 MEDICAL EXPERTS ASSESSMENT

While the generated MRIs preserve all visual details and other essential features, it is not clear how physicians can benefit from the generated MRIs or whether they can clearly distinguish artifacts of the generated MRIs. To test this we conduct a series of small retrospective clinical studies on the generated MRIs of both the brain and knees from our X-Diffusion and with the help of expert physicians test the samples against real MRI samples (summarized in Table 2).

**Small brain MRI clinical study.** We gave a certified neuro-oncologist *W. S.* a set of 20 Brain MRI samples that have both the generated MRIs and the true MRIs as unordered randomized pairs. We asked him to give his decision on which of the samples were the true ones and his precision was only 40 %. This means that the generated brain MRIs are indistinguishable from the real ones even for an expert oncologist. On another test, we asked him to identify if the generated MRIs on another 10 samples have enough tumour information and rate them from 1 to 10, where 10 means all information about the tumour is present, clear, and realistic. The score of the generated tumour was $8.6 \pm 1.0$ out of 10.

**Small Knee MRIs clinical study.** To qualitatively assess how realistic our generated knee out-of-domain 3D volumes were (produced from a single slice), we gave 20 generated examples alongside their real MRI counterparts to an expert orthopedic surgeon *J. F.*. He was then asked to identify the real example from a set of 20 MRI pairs. The surgeon correctly identified the real MRI in *only*

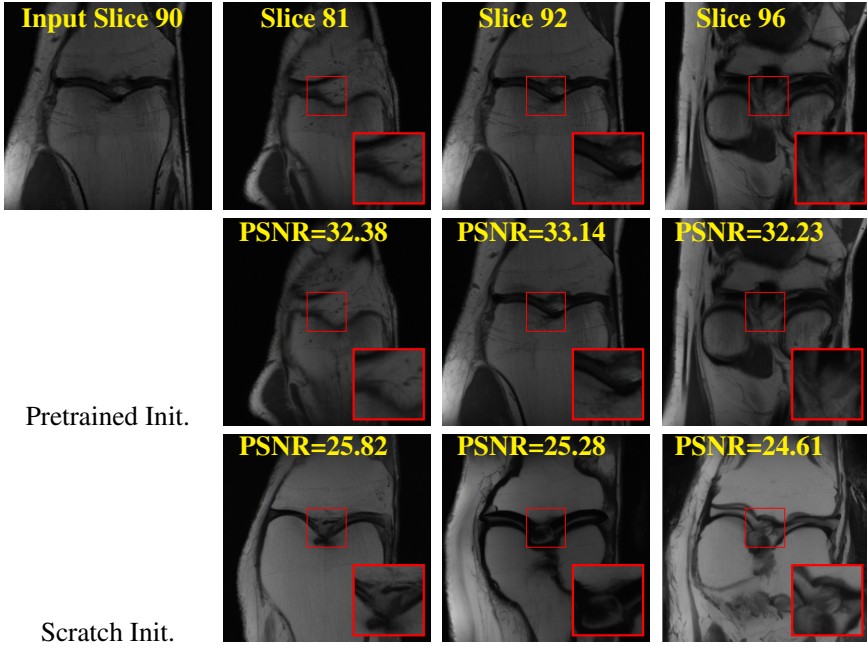

Figure 8: **Out-of-Domain Generations of X-Diffusion.** We show an example of knee 3D MRI generation using X-Diffusion from the *single input slice* on the left. We show (*top*): ground truth slices of the same sample as a reference, (*middle*): generated slices of 3D MRI by X-Diffusion, (*bottom*): the generated slices when X-Diffusion is trained from scratch. Our X-Diffusion can generate high-fidelity 3D MRIs of knees, even though it is trained on BRATS brain MRI dataset, illustrating its potential as a foundation model for 3D MRI generation. The pretraining of X-Diffusion by Stable Diffusion (Rombach et al., 2022a) and Zero-123 (Liu et al., 2023) (same U-Net architecture) helps in the domain generalization, explaining the success of the full X-Diffusion (middle row).

*10* out of 20 pairs, could not decide in 3 pairs, and misidentified the generated MRI as real in the remaining 7 pairs. This further validates the generated out-of-domain MRIs.

## 6 ANALYSIS AND ABLATION STUDY

### 6.1 VOLUME AVERAGING

We study the effect of volume averaging at inference as detailed in Section 3.2. We note (from Figure 7) how the averaging volumes indeed increase the performance up to a certain point. The results of 3D PSNR (dB) for the 31-slices X-Diffusion on $N = 1$, 2, 3, 5, and 10 volumes are 35.48, 35.94, 36.17, 37.40, and 36.72 respectively. This is consistent with multi-view understanding literature when the number of views increases, performance generally increases (Hamdi et al., 2021).

### 6.2 ORTHOGONAL SLICES INPUT

All of the results of multi-slice input of X-Diffusion shown in Section 5 are from input slices of the same axis (sagittal, coronal, or axial). It might be intriguing to see what if the input multi-slices were from different orthogonal planes, the results would improve over the ones shown in Table 1. For this, we train on the Brats dataset from orthogonal planes a 2-slice X-Diffusion (coronal + sagittal) and a 3-slice X-Diffusion and the test PSNR results would be 25.61 dB and 29.77 dB respectively. This is compared to 25.20 dB and 29.43 dB for same-axis 2-slices and 3-slices results of Table 1.

### 6.3 WHY DOES X-DIFFUSION WORK?

**The Effect of Pretraining.** We hypothesize that the massive pretraining of our X-Diffusion based on Stable Diffusion weights (Rombach et al., 2022a) played an important role. Another aspect is that the Zero-123 (Liu et al., 2023) weights which are modified Stable Diffusion weights that understand

| Body Part MRI | Sample | Real MRI Detection Rate | Average Pathology Grade |
|---|---|---|---|
| Brain | 20 | 40.0% | 8.6/10 |
| Knee | 20 | 58.8% | N/A |

Table 2: **Medical Experts Assessment of Brain and Knee MRIs.** For a set of twenty brain and knee MRIs, the detection rates for the real MRI from randomized pairs of real and generated MRIs are almost 50%, indicating that the generated samples are indisputable from real MRI samples. On a separate set of ten generated brain MRIs, tumour information was assessed by a neuro-oncologist on a scale of 1 to 10, where 10 means all information about the tumour is present, clear, and realistic.

viewpoints and fine-tuned on large 3D CAD dataset Objaverse (Deitke et al., 2023) can indeed be the reason why X-Diffusion generalizes well. The PSNR for 1-slice on BRATS dataset are (SD-pretraining): 21.52 dB, (Zero-123-pretraining): 23.13 dB, (no-pretraining): 17.14 dB. These results highlight the importance of pertaining to X-Diffusion. Refer to Figure 8 for similar observation.

**Leveraging Context.** Since we train on a cancerous brain dataset, one question that might arise is whether X-Diffusion generated brain MRIs preserve tumour information when the given inputs do not intersect with any tumour. We perform experiments varying the input slice index for 3D brain MRI generation. We measure the performance for input slices with no intersection with the tumour (not a single pixel with tumor). We also measure performance when only input slices are selected from tumor range. The Dice Scores of the random slices, no-tumour, and only-tumour are 83.09, 79.23, and 83.68 respectively. As can be seen here, the brain volumes generated from input slices with no tumour still preserve tumour information despite a small drop in performance. This indicates that X-Diffusion *is* leveraging the context to preserve key information, such as tumor locations. This observation is consistent with how tumor segmentation models with global context (Cao et al., 2022) perform better than local-based U-Nets. More details are provided in *Appendix*.

### 6.4 WHEN DOES X-DIFFUSION FAIL?

To see when and how X-Diffusion fails, we conducted an experiment on healthy brains (no tumour) using the IXI dataset, by running an X-diffusion trained on the BRATS brain tumor dataset. Our X-Diffusion achieved a PSNR of 35.86 dB on the IXI dataset despite being trained on the BRATS dataset. We then ran the tumour segmenter on the set of 582 healthy scans and corresponding generated MRIs. The segmenter predicted tumours in 9.9% of the real healthy brains and in 11.3% of the generated brain MRIs. Some of these tumor hallucination examples from X-Diffusion generation are shown in Figure 9.

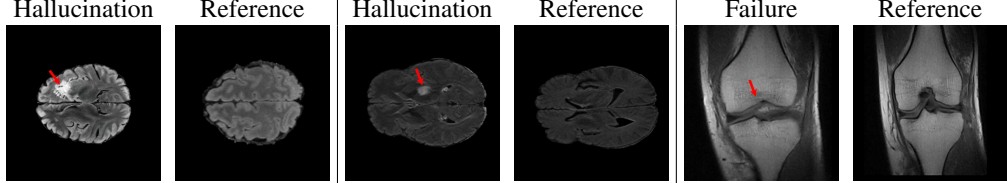

Figure 9: **Tumour Hallucination and Failure Cases in X-Diffusion Generation.** We show two cases of failure (*red* arrow) of our model hallucinating tumour in healthy sample scans. These tumour hallucinations represent only 2% of the healthy test set. Also, we show a failure case for the out-of-domain knee generation with the reference ground truth MRI slice.

## 7 CONCLUSIONS AND FUTURE WORK

*X-Diffusion* achieves high precision with limited inputs, as confirmed by tests on BRATS and UK Biobank data. Future directions include extending its application to dynamic MRI types and exploring its utility in other domains like environmental sciences.

**Limitations.** *X-Diffusion* occasionally exhibits minor artifacts in complex tissue interfaces, a known issue in generative models operating in input-sparse scenarios. An instance of this is discussed in Section 6.4 and *Appendix* with additional examples. Additionally, the high-quality images generated by X-Diffusion (confirmed by physicians) do not necessarily imply that the generated MRIs contain accurate diagnostic information. Future clinical validation is required to determine their suitability for diagnostic purposes.

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

APPENDIX

## A DETAILED SETUP

### A.1 DATASETS

We conducted our experiments on two primary datasets:

**BRATS.** The largest public dataset of brain tumours consisting of 5,880 MRI scans from 1,470 brain diffuse glioma patients, and corresponding annotations of tumours(Baid et al., 2021; Menze et al., 2015a; Bakas et al., 2017). All scans were skull-stripped and resampled to 1 mm isotropic resolution. All images have resolution $240 \times 240 \times 155$, and we use the flair T2 sequence. Tumours are annotated by expert clinicians for three classes: Whole Tumour (WT), Tumour Core (TC), and Enhanced Tumour Core (ET). We split the 5,880 MRIs split into Train (n=4704), Validation (n=588), and Test (n=588) sets.

**UK Biobank.** A more comprehensive dataset of 48,384 full-body MRIs from more than 500,000 volunteers(Sudlow et al., 2015). UK Biobank MRIs are resampled to be isotropic and cropped to a consistent resolution ($501 \times 160 \times 224$). 48,384 whole-body MRIs are paired with antero-posterior (AP) DXA scans of the same subjects. These Dixon MRIs do not come stitched, the scans are scanned axially and there is a disparity in the bias field effect (a common artifact of MRI machines) which is strongest at the knee region. These Dixon MRI patches could not be stitched seamlessly with our current pipeline. These artifacts appear on all scans of the UKBiobank that we stitch. Therefore, the X-Diffusion trained on this data will recreate these artifacts regardless of input. The same pattern is present on all samples in the dataset for a fixed depth, while different depth indices will have different fixed patterns. We made sure there was a coherence split, such that each patient was in a unique set. We will publish the unique IDs used for train-validation-testing to confirm there is no leakage, nor retrieval of images. Both datasets are pre-processed to ensure compatibility with the X-Diffusion pipeline and to maximize the fidelity of the generated results. Pre-processing includes data normalization to the range [0,1], conversion to fit the RGB channel expected from the pre-trained diffusion model via replicating the grayscale to each channel, and padding to fit network input resolution 256x256x3.

For Validation experiments, we use the following datasets:

**IXI.** It is a dataset of T1-weighted MR images of 582 healthy subjects, freely available online (IXI). IXI dataset was collected from three different hospitals in London: Hammersmith Hospital using a Philips 3T system, Guy's Hospital using a Philips 1.5T system and the Institute of Psychiatry using a GE 1.5T system.

**Knee fastMRI.** It is a public dataset of raw k-space data from NYU Langone(Knoll et al., 2020; Zbontar et al., 2019). We use the test set provided (n=109) of fastMRI single coil, of dimension 640x372x30. The knee MRIs are center-cropped to 320x320x30.

**Synthetic Volumes.** It is our own generated synthetic volumes of 3D cones with different colors, sizes, and orientations (see Figure XII for multiple examples). It has 10K samples to see if X-Diffusion can learn 3D volumes other than MRIs. We varied the parameters for the inner volume of the cone and color gradient to generate cones with varying sizes and colors. The dataset is split into 80% for training (8k), 10% for validation (1k), and 10% for testing (1k).

### A.2 EVALUATION METRICS

To quantify the efficacy of X-Diffusion, we employed a suite of evaluation metrics, namely:

- **Peak Signal-to-Noise Ratio (PSNR)**: Indicates the quality of the reconstructed MRI by assessing the fidelity of the generated MRI in relation to the original.

$$PSNR(x, \hat{x}) = 10 log_{10}\left(\frac{max(x)^2}{\frac{1}{n}\sum_{i,j,k}(x_{i,j,k} - \hat{x}_{i,j,k})^2}\right) \quad (4)$$

  where x represents the ground truth volume, $\hat{x}$ is the predicted volume, and n is the total number of voxels in the ground truth volume.

- **Structural Similarity Index (SSIM)**: Captures the perceived changes between the original and generated MRI images.

$$SSIM(x,\hat{x}) = \frac{(2\mu_x\mu_{\hat{x}} + C_1) + (2\sigma_{x\hat{x}} + C_2)}{(\mu_x^2 + \mu_{\hat{x}}^2 + C_1)(\sigma_x^2 + \sigma_{\hat{x}}^2 + C_2)} \tag{5}$$

where $x$ denotes the ground truth slice, $\hat{x}$ is the predicted slice, $\mu x$ is the average of $x$, $\sigma_x^2$ is the variance of $x$, $\sigma_{x\hat{x}}$ is the covariance between $x$ and $\hat{x}$, $C_1=(k_1 L)^2$, $C_2=(k_2 L)^2$, L is the dynamic range of pixel values, and $k_1$=0.01 and $k_2$=0.03.

We measured the random PSNR on the whole test set for reference on the UKBiobank, BRATS, and knee fastMRI dataset. For the UKBiobank, two randomly sampled MRIs have a PSNR of 15.95 ± 0.36 dB. For BRATS, it is of 19.89 ± 1.59 dB, and for the knee fastMRI of 20.21 ± 2.58 dB.

**On BRATS dataset only**

- **Dice Score:** We use the average Dice score to evaluate the performance of our model at segmenting the brain tumours (Menze et al., 2015b): Dice Score = $\frac{2|Y \cap \hat{Y}|}{D(|Y|+|\hat{Y}|)}$, where $Y$ is the prediction, $\hat{Y}$ is the ground-truth label and $D$ the total number of slices in the set.

- **Brain Volume:**
We measure brain volume in $mm^3$ by counting the non-zero voxels in the volume multiplied by the volume in $mm^3$ of each voxel (Dikici et al., 2019).

$$NonZeroVoxCount = \sum_i^N V(x_i, y_i, z_i) > 0 \tag{6}$$

$$VoxVol(mm^3) = v_x * v_y * v_z \tag{7}$$
$$BrainVol = NonZeroVoxCount * VoxVol$$

**On UK Biobank dataset only**

- **Ground-truth Correlation Index:** Pearson's correlation coefficient $r$ measures the strength of a linear association between two variables. The formula in 8 returns a value between -1 and 1, where: 1 denotes a strong positive relationship; -1 denotes a strong negative relationship; and zero denotes no relationship (Bourigault et al., 2022).

$$r = \frac{\sum_{i=1}^{n}(x_i - \bar{x})(y_i - \bar{y})}{\sqrt{\sum_{i=1}^{n}(x_i - \bar{x})^2}\sqrt{\sum_{i=1}^{n}(y_i - \bar{y})^2}} \tag{8}$$

- **Spine Curvature** Let $\gamma(t) = (x(t), y(t))$ be the equation of a twice differentiable plane curve parametrized by $t \in [0, 209]$. We measure the spine curvature $\kappa$ with the standard mathematical formula (Bourigault et al., 2023):
$\kappa = (y''x' - x''y')/(x'^2 + y'^2)^{\frac{3}{2}}$.

## A.3 IMPLEMENTATION DETAILS

We implement X-Diffusion based on the Stable Diffusion (Rombach et al., 2022a) U-Net with additional controls and conditions. We detail some of the hyperparameters and design choices below.

For the first stage of autoencoder training, the encoder downsamples the image $x \in R^{H \times W \times 3}$, where $H = W = 256$ by a factor 8 to allow the DPM to focus on the semantic features of the latent space in a computationally efficient manner. KL regularization is added to mitigate high variance latent space. In the second stage, a DPM is trained on the learned lower-dimensional latent space. The configuration of the U-Net is as follows: 2 residual blocks, channels multiples: [ 1, 2, 4, 4 ], attention resolutions: [ 4, 2, 1 ], 8 heads, using a spatial transformer with depth = 1. For the DDPM Latent Diffusion, we use a base learning rate of $1.0^{-06}$, timesteps $T = 1000$, image size = 32, channels = 4, and hybrid conditioning (concatenation and cross attention). Sampling is performed with classifier-free guidance (see Figure3 for an example of test time sampling).

We use image-conditioned stable diffusion $v2$ checkpoint from Lambda Labs. We follow the novel view synthesis training from Zero-123. X-Diffusion is trained on a single GPU a6000, 48GB of RAM for four days.

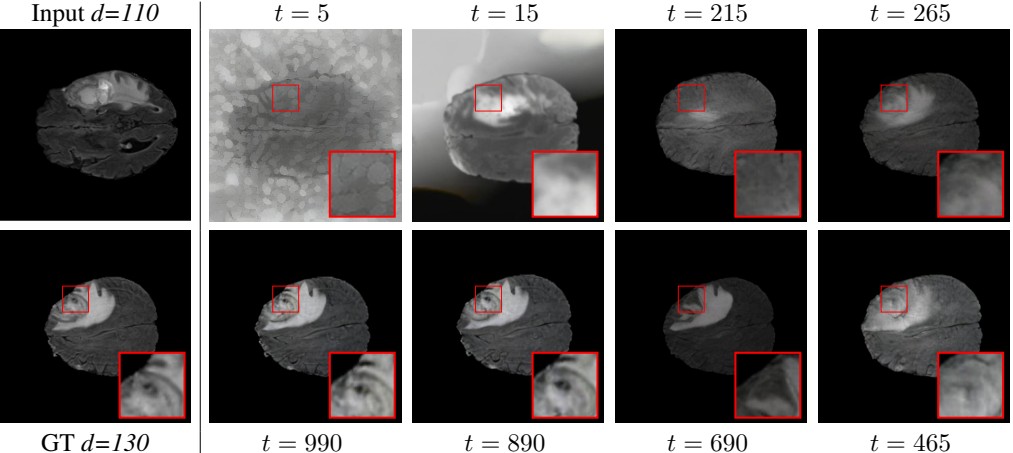

Figure I: **Test Time Brain Generation at Different Sampling Steps**. For the input slice 107 (*top left*), we show the ground-truth slice 90 (*bottom*) and corresponding brain slice generating at different sampling steps $t$ in the denoising diffusion process.

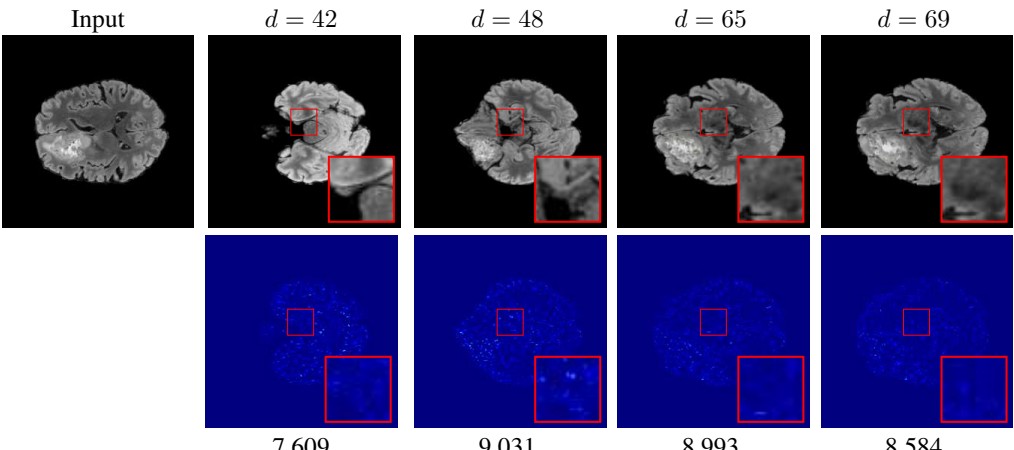

Figure II: **Residual Error of Generated MRIs**. For the input slice (*left*), we show a difference map(*bottom*) between generated MRI (*top*) and ground truth. Below the (*bottom*) row, we indicate the mean squared error between generated and ground-truth images. Brighter pixels indicate greater disparity.

$d = 97$      $d = 104$      $d = 108$      3D Tumour

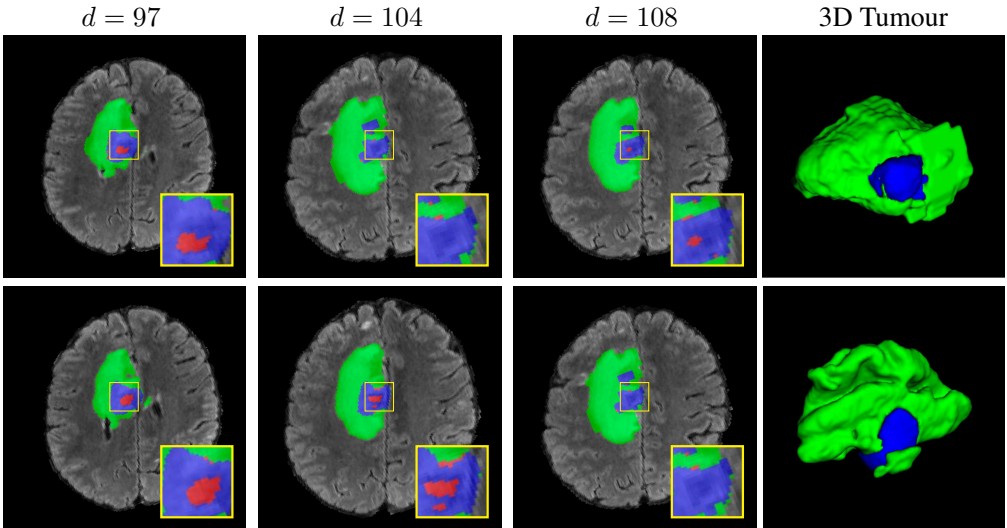

Figure III: **Generated MRIs with Segmentation Maps Overlaid**. We show ground-truth segmentation maps(*bottom*) and generated MRI (*top*). Red is used for the non-enhancing and necrotic tumor core, green for the peritumoral edema, and blue for the enhancing tumor core. The 3D Dice Score for this example is 77.26.

Input

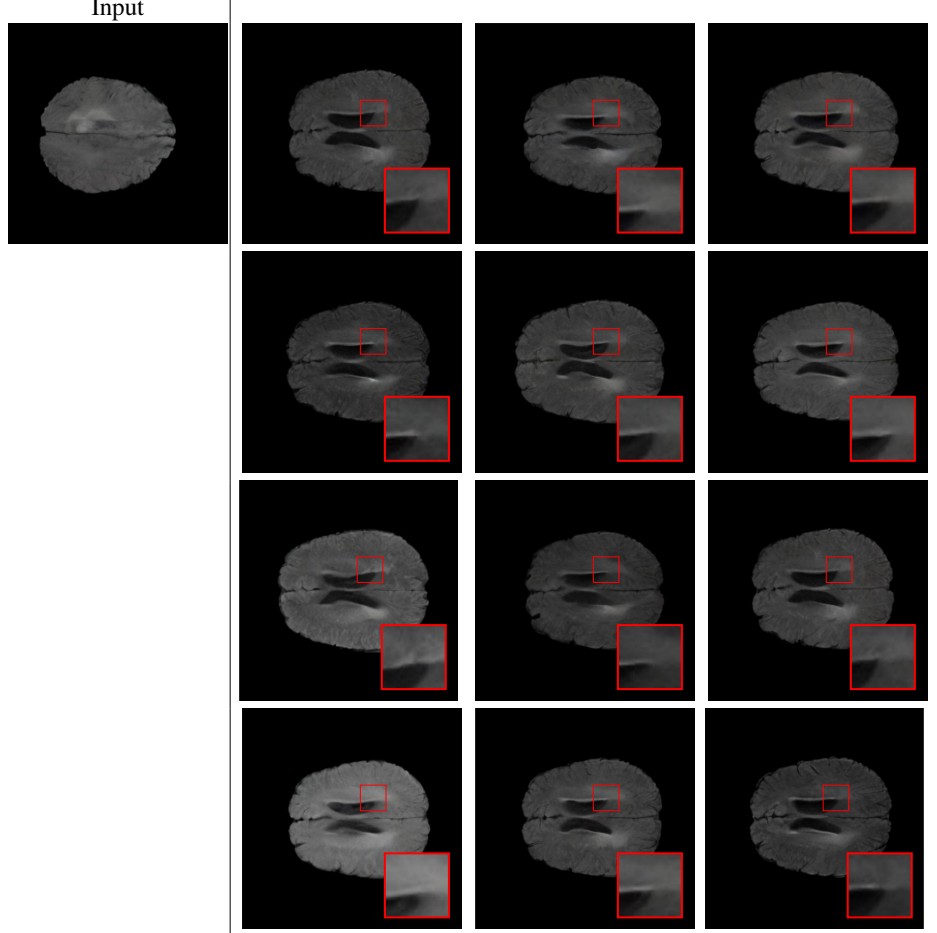

Figure IV: **Probabilistic Output for Different Volumes Generated from Single Slice by X-Diffusion**. For the same input slice (*top left*), we show 12 generated output slices ( at index $d = 88$) using 12 different inputs Gaussian noise for X-Diffusion U-Net.

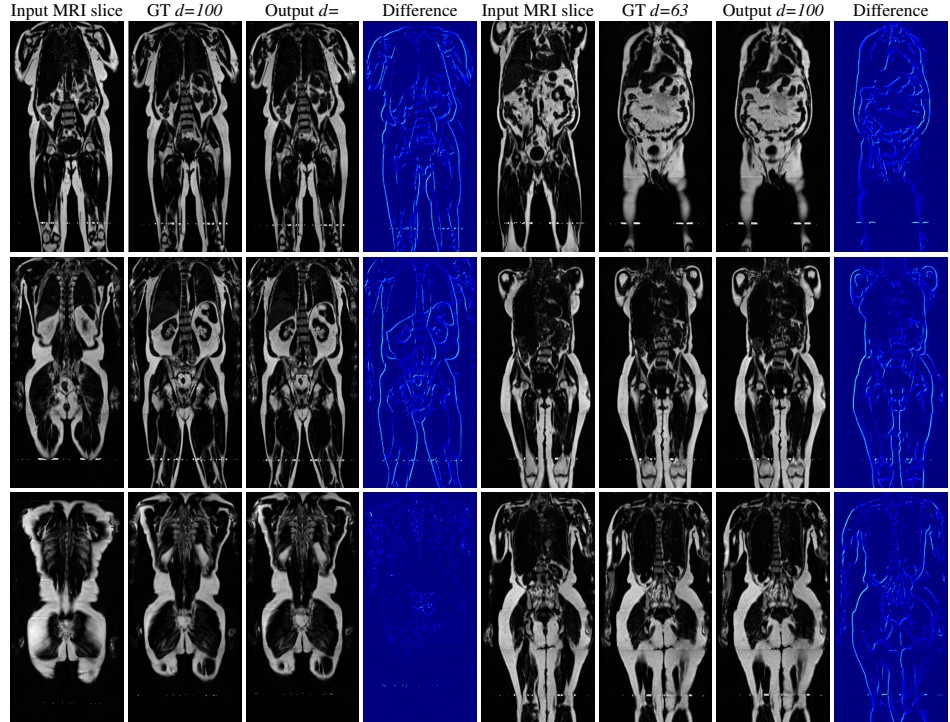

Figure V: **Qualitative Results of full body 3D MRI Generation with X-Diffusion.** We show a single MRI slice example, Two corresponding ground-truth MRI slices (index 63 and 100), the corresponding generated MRI slice, and a difference map to qualitatively measure the error between generated and ground-truth MRI. Note that when stitching the MRIs in the UKBioBank dataset, there is a disparity in the bias field effect which is strongest at the knee region (brighter pixels). The same pattern is present on all samples in the dataset for a fixed depth, and hence it is learned by X-Diffusion as well.

# B    ADDITIONAL RESULTS

## B.1    FAT VALIDATION

We ran further experiments to investigate whether the generated MRIs (see an example in Figure V) preserve fat information. We use an image-based regression network trained on the UKBiobank to estimate DXA metadata information from 2D compressed middle coronal and sagittal MRIs (Langner et al., 2021). Pearson's correlations using Equation 8 comparing reference values and generated values are reported in Figure IX with most fields having high correlation $r > 0.9$. We show that the generated MRIs preserve crucial internal information.

## B.2    BRAIN VOLUMES PRESERVATION

The comparison of generated MRIs versus reference MRIs suggests a nearly perfect preservation of brain volume (in mm³) with median volume of reference MRIs of $1.31e^6\ mm^3$ versus generated MRIs $1.28e^6\ mm^3$ (see an example of brain generation in Figure II).

## B.3    PRESERVATION OF SPINE CURVATURE AND FAT

For the spine segmentation on UK Biobank, we use a UNet++ model (Zhou et al., 2018) with Dice Loss. We use a model trained to predict curves on DXA on UK Biobank (Bourigault et al., 2022). We show in Figure XIV that generated MRIs preserve the spine curvature from normal to severe scoliosis cases. We also study the case when DXA is used to generate the MRIs and show in Figure VI how the correlation to real curvatures compares to the input MRI case. The curvatures of the MRI generated from the coronal plane match the DXA curvatures more than the curvatures generated from sagittal

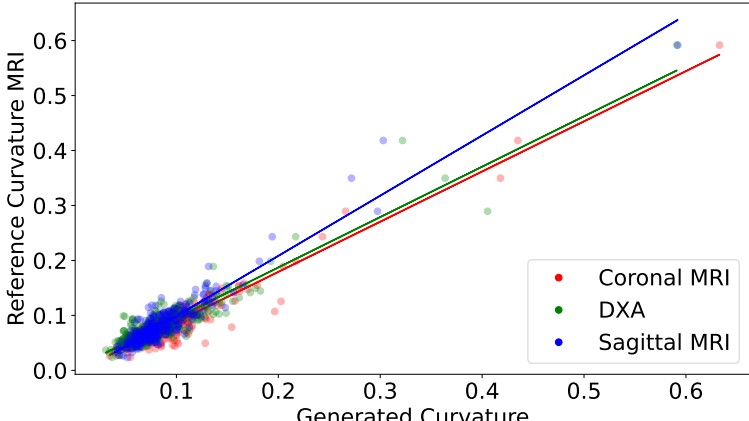

Figure VI: **Curvature Preservation of Generated MRIs.** We plot spine curvature measured on reconstructed MRIs where the input was either, (i) a single MRI coronal slice, (ii) a single sagittal slice, or (iii) from the paired DXA, against the curvature of reference real MRIs of the same samples. The correlation coefficients are 0.89 for the MRIs, 0.88 for the MRIs from sagittal plane generation, and 0.87 for the DXAs.

MRI. This is expected since the antero-posterior plane of DXA is equivalent to the coronal plane for MRIs. This also explains the greater Pearson's correlation coefficient $r$ of the coronal MRI (0.89) and DXA-generated curvature (0.88) compared to sagittal-generated curvature (0.87) relative to the reference curvature on the coronal plane. We observe though that MRI generation using X-Diffusion from another plane than the conventional plane for scoliosis assessment is valid.

### B.4 TUMOUR INFORMATION PRESERVATION

On the test set with human ground-truth annotations ($n = 333$), the brain volumes generated from single slice input preserve the volume of the different tumour components (paired t-test, $p-value < 0.05$ for all 3 classes) (see Table I). The real MRI Dice scores are put for reference to our generated MRIs. X-Diffusion outperforms baselines TPDM (Lee et al., 2023) and ScoreMRI (Chung & Ye, 2022) in tumour preservation (see Table I and Figure III). We ran experiments comparing the tumour segmentation Dice Score varying X-Diffusion configurations. The multi-slice input X-Diffusion achieves a marginally better Dice Score than the single-slice input model ($83.47 \rightarrow 83.09$). We also ran experiments with slice input used for volume reconstruction intersecting or not with tumour. We observe on average a drop of 6% Dice Score (see Table I). Further away from the tumour the input slice for volume reconstruction is selected, and we observe a linear decrease in tumour segmentation Dice Score with the lowest value of 77.21 Dice Score (see Figure VII).

This shows how the generated MRIs indeed preserve the tumour information and can act as an affordable and informative pseudo-MRI, before conducting an actual costly MRI examination in hospitals. Given that our model has been trained on brain scans all with tumours, we expect to see hallucinations of tumours in healthy scans. We report two cases of failure of our model in Figure VIII. Hallucinations of tumours on healthy samples represent 2% of the test set.

## C ADDITIONAL ANALYSIS

### C.1 ABLATION STUDY

**Repeated Input Single Slice in Multi-Slice Models.** We try to see whether the multi-slice models are better than single-slice models by studying if we used repeated input single slice multiple times. The 3D PSNR results for multi-slice input with 1, 2, 3, 5, 10, 31, and 60 repeated slices are 23.1, 23.256, 23.638, 23.921, 24.379, 25.125, and 24.921 respectively.

| X-Diffusion Generated MRIs | Test Dice Score ↑ | | | | |
| --- | --- | --- | --- | --- | --- |
| | ET | WT | TC | Average Dice | 3D PSNR(dB)↑ |
| single slice | 75.48 | 89.24 | 84.57 | 83.09 | 35.81 |
| multi-slice | 75.82 | 89.56 | 85.04 | 83.47 | 36.13 |
| multi-slice (only-tumour) | 76.12 | 90.04 | 85.87 | 84.01 | 36.98 |
| multi-slice (no-tumour) | 70.14 | 84.29 | 81.65 | 78.69 | 33.24 |
| Real | 76.47 | 91.13 | 86.24 | 85.15 | N/A |

Table I: **Dice Score for Brain Tumor Segmentation on Real MRI vs. Reconstructed MRI.**. We show Dice Score for generated MRIs ($n = 587$ test samples) by our X-Diffusion when input only intersection with tumour (only-tumour) and when input does not intersect with tumour (no-tumour) for a single slice and multi-slice input (31 slices). Note how X-Diffusion predicts the correct 3D tumour locations even when the input 2D slice does not intersect the tumour in most cases (drop from 83.47 to 81.65 Dice Score). ET: Enhancing Tumor, WT: Whole Tumor, TC: Tumor Core.

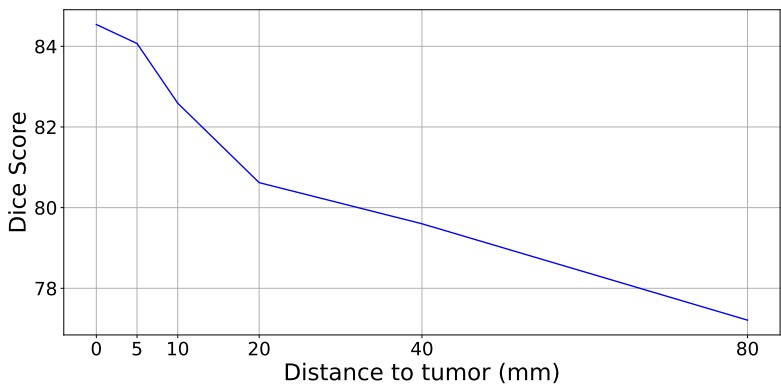

Figure VII: **Dice Score versus Distance to Tumour.** We show the decrease in Dice Score for slice selection at increasing distances from the center of the tumor. This distance goes up to $80mm$ (where slice index $\in [1, 5] \cup [151, 155]$, total number of slices is 155 per scan, and $n = 587$ test samples). These results indicate that the proximity of input slices to the tumor significantly impacts reconstruction accuracy.

| Failure | Reference | Failure | Reference |
| --- | --- | --- | --- |

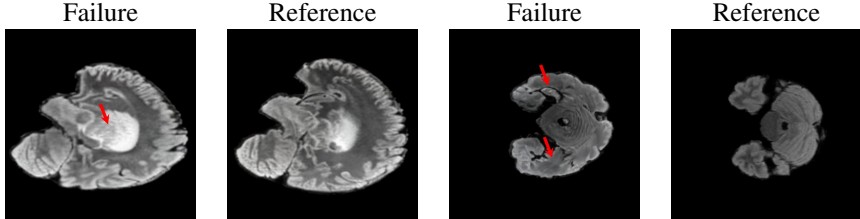

Figure VIII: **X-Diffusion Failure Cases.** We show two cases of failure (*red* arrow) on BRATS generations.

| Method | PSNR↑ | SSIM↑ | | |
|---|---|---|---|---|
| | | Axial | Coronal | Sagittal |
| X-Diffusion (single) | 34.17 | 0.88 | 0.87 | 0.88 |
| X-Diffusion (multi) | 36.57 | 0.89 | 0.88 | 0.89 |

Table II: **Out-of-Domain Generalization**. We evaluate 3D knee generation using 3D PSNR and mean SSIM on the test set of $n = 109$ knees from NYU fastMRI (Knoll et al., 2020; Zbontar et al., 2019). X-Diffusion is trained on *brain* MRIs from BRATS.

| | Test 3D PSNR ↑ | | | | | | | |
|---|---|---|---|---|---|---|---|---|
| Input Slices | 1 slice | 2 slices | 3 | 5 | 10 | 31 | 60 | 120 |
| **X-Diffusion** | 22.30 | 23.50 | 24.63 | 25.77 | 26.79 | 25.55 | 24.44 | 24.24 |

Table III: **Reconstruction Quality on the Synthetic Cone Dataset**. We report the test 3D PSNR on synthetic volume generation of our model X-Diffusion for varying input slice numbers in training. The synthetic cone dataset is described in Section A.1.

**The Effect of Pretraining.** We hypothesize that the massive pretraining of our X-Diffusion based on Stable Diffusion weights (Rombach et al., 2022a) played an important role. Another aspect is that the Zero-123 (Liu et al., 2023) weights which are modified Stable Diffusion weights that understand viewpoints and fine-tuned on large 3D CAD dataset Objaverse (Deitke et al., 2023) can indeed be the reason why X-Diffusion generalizes well to out-of-domain dataset (see generalization to knee MRIs in Figure XI).

We show the results in the following Table IV.

**Different Mechanisms for Multi-Volume Aggregation.** We used view-dependent volume averaging as described in the main paper in all of the main results in the work. We show probabilistic outputs in Figure IV for different brain volumes generated for a single slice. We show the results of varying the number of volumes in Table VIII. We see that as the number of volumes averaged increases, the performance increases up to a certain point before saturating (as noted in the multi-view literature (Hamdi et al., 2021)). We did try to use other ways to aggregate the view-dependant volumes (*eg.* by max pooling the volumes) and show the results also in Table VIII.

**MRI Volumes Specificity.** One hypothesis that can justify why the X-Diffusion model works very well on MRIs is that MRI data is not ordinary volume data since it is obtained by actually running an inverse Fourier transform on different k-frequency components, which means that the 3D information is embedded in every slice of the MRI. Introducing this Fourier effect on our synthetic Cone volumes dataset by applying masks on the high frequencies and then inverse Fourier results in a slight improvement of volume reconstruction of $+0.51$ PSNR (dB) higher than with no Fourier masking ($26.788 \rightarrow 27.298$ dB). This indicates that the Fourier frequency effect is negligible and does not explain away the performance of X-Diffusion.

## C.2 TIME AND MEMORY REQUIREMENTS

Lowering reconstruction speed is important for greater accessibility, MRI re-acquisition purposes, and to monitor surgery in the case of dynamic MRI. The number of model parameters should be kept low to enable implementation on machines with lower memory capacity. X-Diffusion is on par with other diffusion-based baseline models, albeit higher in memory requirements than classical methods. However, X-Diffusion is the only 3D medical imaging diffusion model that shows the

| Models | 3D PSNR↑ | | | | | | |
|---|---|---|---|---|---|---|---|
| | 1 slice | 2 slices | 3 slices | 5 slices | 10 slices | 31 slices | 60 slices |
| X-Diffusion (pre-training) | 23.13 | 25.25 | 29.43 | 31.25 | 33.27 | 35.48 | 33.18 |
| X-Diffusion (no-pretraining) | 21.52 | 23.42 | 25.16 | 27.06 | 29.32 | 27.86 | 27.43 |

Table IV: **X-Diffusion with Pre-Training versus no Pre-Training**. We show a comparison of X-Diffusion with fine-tuning pre-trained Stable Diffusion weights versus no pre-training.

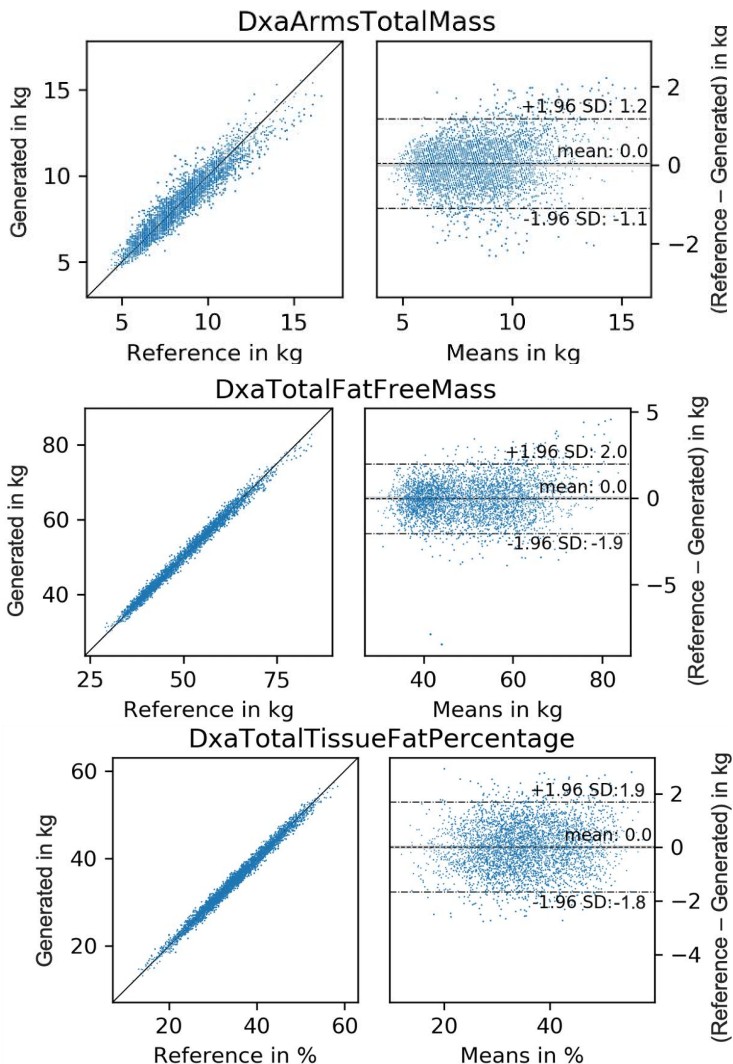

Figure IX: **Evaluation of Body Composition Metrics for Reference versus Generated MRIs**. Each row shows the correlation (*left*) and Bland-Altman plot (*right*) for a different metric: **(Top)** Arm Total Mass (DxaArmsTotalMass), **(Middle)** Total Fat-Free Mass (DxaTotalFatFreeMass), and **(Bottom)** Total Tissue Fat Percentage (DxaTotalTissueFatPercentage). Strong correlations are observed for DxaTotalFatFreeMass and DxaTotalTissueFatPercentage ($r > 0.95$), while DxaArmsTotalMass has a slightly lower correlation ($r < 0.95$).

capacity to generalize beyond the training data, opening the potential for foundation models in 3D MRIs. We show the cost analysis in Table VI.

| Models | Test 3D PSNR ↑ | | | | | | | |
| --- | --- | --- | --- | --- | --- | --- | --- | --- |
| | 1 vol | 2 vol | 3 vol | 5 vol | 10 vol | 20 vol | 31 vol | 60 vol |
| X-Diffusion (max-pool) | 35.48 | 35.48 | 35.52 | 35.48 | 35.31 | 35.46 | 35.19 | 35.33 |
| X-Diffusion (averaging) | **35.48** | **35.94** | **36.17** | **37.40** | **36.72** | **36.35** | **36.83** | **36.53** |

Table V: **Effect of Volume Averaging on The Performance**. We show best performing model (31 slices) on BRATS with number of volumes averaged from view-dependent 3D MRI generation. We see that the PSNR reaches a peak for 5 volumes averaged before stabilising at 10 volumes. We also compare with a variant that takes the maximum of volumes instead of averaging.

| Models | #Params | Runtime (s) |
|---|---|---|
| Score-MRI(Chung & Ye, 2022) | 860M | 139.142 |
| TDPM(Lee et al., 2023) | 1720M | 149.468 |
| X-Diffusion | 990M | 141.461 |

Table VI: **Cost Analysis.** We show compute cost and runtimes that are measured on a computer with a single GPU a6000, 48GB of RAM.

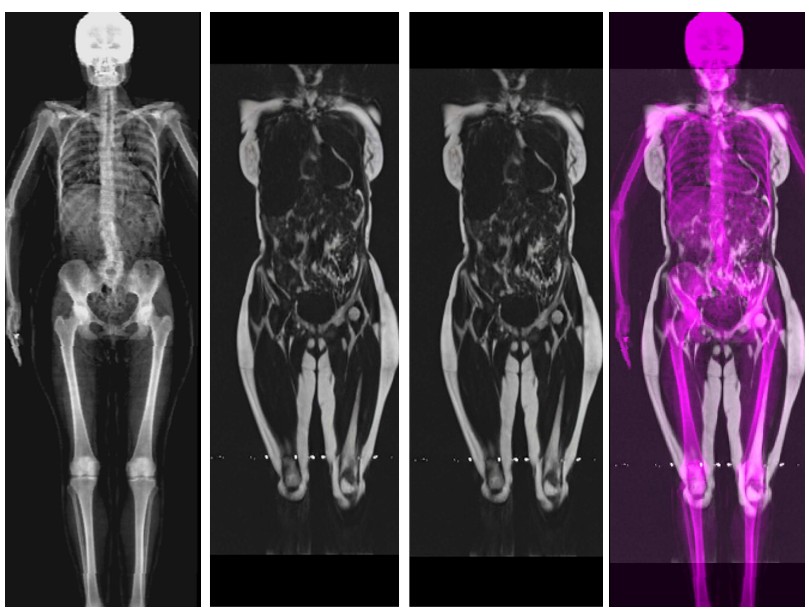

Figure X: **Application of X-Diffusion: DXA to MRI Generation**. We show an example of applying X-Diffusion on DXA-to-MRI generation. From left to right: input DXA, ground-truth MRI, generated MRI, and overlay of the generated MRI and the input DXA to test the alignment. The 3D PSNR for this example is 26.38 dB.

### C.3 COMPRESSED SENSING EXPERIMENT

Some of the previous works on MRI reconstruction (Chung et al., 2023; Chung & Ye, 2022) target the task of compressive sensing, where the goal is to increase the frequency resolution of the MRIs when the k-space is undersampled. While this is not the goal of X-Diffusion, we adapted X-Diffusion to this task and train X-Diffusion on the k-space of the MRIs. The performance for our model in the compressive sensing task for under-sampling factor $\alpha = 2$ is $PSNR = 35.17$ dB. Results are shown in Table VII.

| Acceleration Factor | Test 3D PSNR ↑ | | |
|---|---|---|---|
| | 2 | 4 | 6 |
| X-Diffusion | 35.17 | 34.41 | 34.16 |
| DiffusionMBIR(Chung et al., 2023) | 37.16 | 36.12 | 35.85 |
| TPDM(Lee et al., 2023) | 36.48 | 35.52 | 35.18 |
| ScoreMRI(Chung & Ye, 2022) | 34.18 | 33.88 | 33.57 |

Table VII: **Compressive Sensing Experiment**. We show test 3D PSNR for benchmark models DiffusionMBIR(Chung et al., 2023), TPDM(Lee et al., 2023), and ScoreMRI(Chung & Ye, 2022), and X-Diffusion for input downsampled by acceleration factor 2, 4, and 6. This shows that X-Diffusion is not perfect for compressive sensing, as its power in the spatial domain.

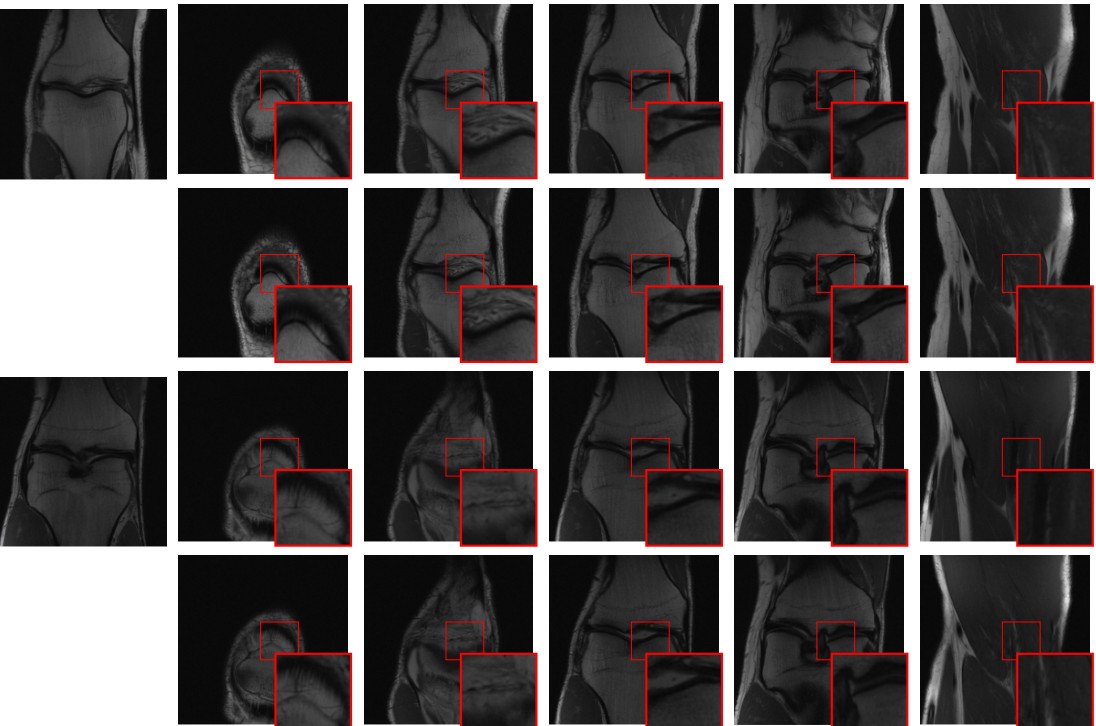

Figure XI: **Out-of-Domain Knee Generations of X-Diffusion 1.** We show two examples of knee 3D MRI generation using X-Diffusion from the *single input slice* on the left. We show (*top*): different slices of the generated 3D MRI, (*bottom*): ground truth slices of the same sample as reference. Mean PSNR for *top* example is of 36.84 dB and for *bottom* example of 35.17 dB.

| Models | Test 3D PSNR ↑ | | | | | |
| --- | --- | --- | --- | --- | --- | --- |
| | 1 slice | 2 slices | 3 slices | 5 slices | 10 slices | 31 slices |
| X-Diffusion (Avg. Dot) | **23.1** | **25.2** | **29.43** | **31.25** | **33.27** | **35.48** |
| X-Diffusion (MLP) | 22.7 | 24.91 | 28.89 | 30.73 | 32.82 | 35.16 |

Table VIII: **Comparing Model Performance of Multi-Input Aggregation Procedure on Brain Data**. We compare the MRI reconstruction for X-Diffusion model for varying aggregation procedure i.e. dot averaging and multi-layer-perceptron (MLP) reduction and for varying input slice numbers. We report the mean 3D test PSNR on BRATS brain dataset. The results show that our aggregation method with dot product averaging increases model performance by a margin compared to MLP reduction method for varying number of input slices despite its simplicity.

### C.4 MULTI-SLICE INPUTS

The multi- slice inputs are sampled from the same axis of rotation during training and testing. To reduce the memory requirement for running the pipeline, the reduction operation of the $K > 1$ input slices $(x_1, x_2, ..., x_K)$ is similar to what is followed in TPDM (Lee et al., 2023) in the conditioning volume, and it can be described as follows: $x = \frac{1}{K-1} \sum_{j=1}^{K} x_j \cdot x_{j+1}$. The difference in performance between the simple dot product reduction and the learned reduction with additional MLP is shown in Table VIII. During training, the slices do not need to be consecutive. The diffusion model implicitly learns to handle the slice gap since it is trained on multiple slices with different gaps. For the evaluation of multi-slice benchmarks, fixed input slices are sampled uniformly from the test set and used for all the compared models.

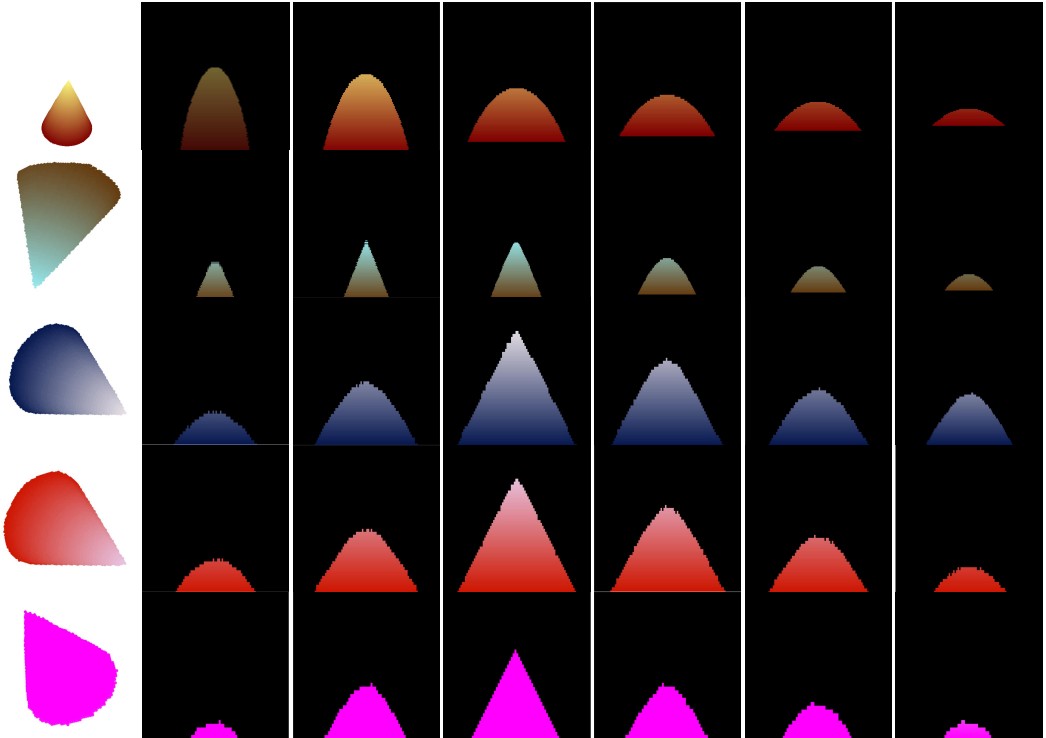

Figure XII: **Synthetic Volume Dataset**. We show some samples of our proposed Synthetic Volumes dataset. The dataset consists of cones with different sizes, orientations, and colours (constant and gradient colours). The dataset is described in details in Section A.1.

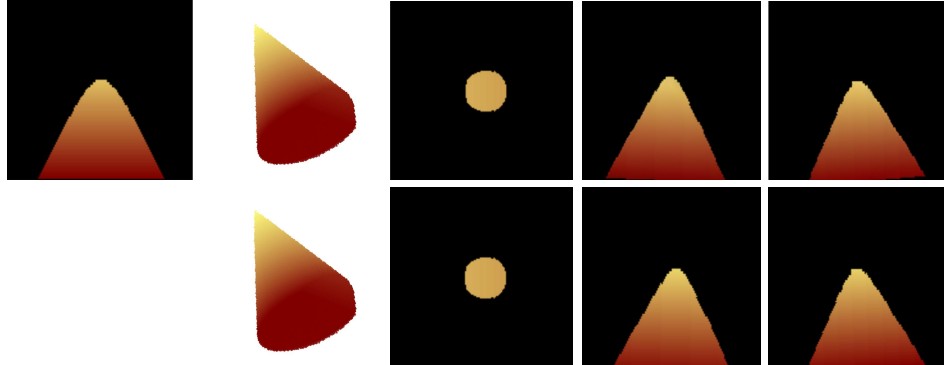

Figure XIII: **Synthetic Cone Generation with X-Diffusion.** Top left image is the input to the model. We show (*top*): the generated 3D cone and slices from the 3D volume, (*bottom*): the ground-truth 3D cone and corresponding slices from 3D volume.

## C.5 SYNTHETIC VOLUMES GENERATION

We applied our X-Diffusion model on a completely different volumetric data modality to see if the MRI volume generation is indeed a simple task for X-Diffusion (see Figure XII). To do this, we trained X-Diffusion on a synthetic volumes dataset. We show an example of the generated volume used for training and the corresponding prediction in Figure XIII and quantitative results in Table III.

## D    ADDITIONAL DISCUSSIONS (DURING REBUTTALS)

### D.1    CLINICAL RELEVANCE OF X-DIFFUSION

While the generated "pseudo MRIs" from X-Diffusion are not intended to replace comprehensive MRI scans, we believe that our work represents an exploratory step toward novel imaging methodologies that could have future clinical relevance. In current clinical practice, MRI scans are time-consuming and expensive due to the need for acquiring comprehensive volumetric data. The cost of MRI varies considerably, depending on the infrastructure costs, and personal staff Bell (1996); Wald et al. (2020); Arnold et al. (2023). In the UK the cost of performing MRI research in a university teaching hospital is typically in the range of £350–£500 per hour of scanner occupation [1]. We envision that, in the future, technologies like X-Diffusion could be integrated into the MRI workflow to enhance efficiency. For example:

- **Preliminary Assessment**: During the initial phase of an MRI examination, X-Diffusion could generate preliminary 3D reconstructions from a limited number of high-quality 2D slices. This could provide immediate insights into the patient's anatomy, allowing radiologists to identify regions of interest quickly.

- **Adaptive Scanning**: With real-time preliminary reconstructions, technicians could adapt the scanning protocol on-the-fly, focusing on areas that require higher resolution or additional imaging sequences, thereby optimizing scan time and resource utilization.

- **Workflow Efficiency**: By potentially reducing the total scanning time, X-Diffusion could increase patient throughput and reduce waiting times, leading to improved access to MRI services.

- **Cost Reduction**: Shorter scan times and optimized imaging protocols could reduce operational costs for healthcare facilities, making MRI examinations more affordable.

We acknowledge that significant challenges remain before such applications can be realized. The current limitations include ensuring the accuracy and reliability of the generated images, particularly for detecting small lesions or subtle pathological changes that may not be captured from limited input data. Our work is intended as a proof-of-concept to demonstrate the potential capabilities of cross-sectional diffusion models in medical imaging. Further research, clinical validation, and collaboration with healthcare professionals are necessary to assess the feasibility and safety of integrating X-Diffusion into clinical practice. We hope that X-Diffusion will inspire future developments in rapid imaging techniques and contribute to ongoing efforts to enhance the accessibility and efficiency of MRI examinations.

### D.2    PLANNED FUTURE CLINICAL STUDY OF X-DIFFUSION

**Objective.** To see if recent generative AI technologies for MRIs (X-Diffusion) are relevant to knee diagnosis. The goal is to validate our pipeline for reconstructing full MRIs from one/few slices with high precision and evaluate its usefulness for clinical assessment of knees. Given expert grading of how abnormal a knee is, we will compare the score given for degeneration of knees between the two sets of knees with and without AI generated knees.

**Specific Aim.** Grade how abnormal a knee is on an external set of 50 reconstructed MRIs from single slices or two slices each using X-Diffusion. Then, compare the score for knee degeneration between the annotated original set of 50 samples from humans and see how they are correlated. First, experts will grade degenerative knees on MRI. Then we test X-Diffusion on the same sampled graded. Given any single slice(s) from a degenerate knee we generate synthetic knees. Finally, we compare the score given as grading for degeneration of knees between the two sets of knees with and without AI generated knees.

**Hypothesis.** The trained X-Diffusion model for generating MRIs is capable of reconstructing unseen MRIs from one/few slices in the clinic with high precision. The generated MRIs also maintain the diagnosis differentiability that make them useful for physicians in the clinical setup.

---

[1]https://www.bhf.org.uk/-/media/files/for-professionals/research/bhf-clinical-research-imaging-scan-costing-guidelines-october-2022.pdf

**Study Design.** The X-Diffusion model we will use is trained on NYU (Zbontar et al., 2019) dataset and base our model on this dataset of 1,500 Knee MRIs of coronal and sagittal MRIs. The test will involve 50 MRI staples from Oxford hospital. The slices that will be used are the middle slices of either coronal or sagittal of the T2 scans of the MRIs. The type of abnormalities that are investigated are either aging-related knee degeneration or specific pathologies. Knee pathology encompasses conditions such as osteoarthritis, rheumatoid arthritis, meniscal tears, ligament injuries, patellofemoral pain syndrome, bursitis, tendonitis, and gout, all of which can cause pain, inflammation, and functional impairment in the knee.

**Randomization.** We opt for a study setting in a similar fashion as randomised controlled trials as they are proven to be the most reliable way to compare two techniques. We want to make sure the only difference between the two sets of knees with and without AI generated knees is effectively AI related. We make sure the MRIs have been acquired in the same way with the same protocol.

**Process Measures.** The physicians will look at the reports of the original and the generated MRIs and will grade them 1 (poor) to 10 (perfect) based on the following criteria:

- Cartilage Integrity: Evaluate the thickness, smoothness, and presence of any lesions or areas of thinning.
- Meniscus Condition: Assess for tears, degeneration, or displacement of the meniscal tissue.
- Ligament Integrity: Check the anterior cruciate ligament (ACL), posterior cruciate ligament (PCL), medial collateral ligament (MCL), and lateral collateral ligament (LCL) for tears, sprains, or degeneration.
- Bone Marrow: Look for signs of bone marrow edema, bruising, or lesions.
- Synovial Fluid: Assess the amount and condition of synovial fluid, looking for effusion or abnormalities.
- Bony Structures: Examine for bone spurs, cysts, or other bony abnormalities.
- Tendon Condition: Evaluate the condition of tendons around the knee for signs of inflammation or tears.
- Patellofemoral Joint: Assess the alignment, smoothness of the cartilage, and presence of any abnormalities in the patella and its tracking.
- Bursae: Check for inflammation or abnormalities in the bursae around the knee.
- Overall Joint Alignment: Evaluate the alignment of the knee joint, looking for signs of valgus or varus deformity.

**Main Outcome Measures.** The proposed research will provide a clinical assessment of the X-Diffusion technology in knee MRIs . Specifically we will measure the pixel level precision of the generated MRIs compared to the original MRIs in PSNR. Furthermore, we will measure the correlation between physicians' grades on different aspects of the reports on the original 50 knee MRI samples and the grades given to the generated MRIs by the physicians.

**Ethical, Privacy and Safety Considerations.** The testing subjects' identities will not appear on the MRI report and no labor is needed as the researchers are the ones involved in the study. Privacy-sensitive content like faces, biometric details, etc will not appear on the MRIs as they will be anonymized . The knee MRI scans are normal and common MRI scans that do not have any permit requirements or constitute a hazard in the hospital either to the staff, or the patients.

### D.3 NOTE ON THE EVALUATION OF X-DIFFUSION AND THE BASELINES

Evaluating X-Diffusion poses unique challenges due to fundamental differences in input data and reconstruction paradigms compared to traditional MRI reconstruction methods (briefly touched on in Figure 1).

**Differences in Conditioning and Setup.** Traditional methods like ScoreMRI (Chung & Ye, 2022) and TPDM (Lee et al., 2023) reconstruct high-resolution images from degraded inputs such as under-sampled k-space data or low-resolution images. In contrast, X-Diffusion conditions on high-quality 2D slices extracted directly from the ground truth (GT) 3D volumes to infer the missing volumetric information. Additionally, the baselines are created to address the setup where the

**Implications for Evaluation.** This discrepancy makes it difficult to directly compare reconstruction quality. It is challenging to separate the model's capability from the influence of conditioning on accurate GT slices. Standard metrics like PSNR and SSIM may unfairly favor models with more informative inputs. Existing baselines may not perform optimally when adapted to use high-quality conditioning data they were not designed for.

**Addressing the Challenges.** To ensure a fair assessment, we:

- Adapted ScoreMRI Chung & Ye (2022) For comparison with Score-MRI for the number of slices used as input. We uniformly sample n slices along the z-axis i.e 1,2,3,5,10,31 and perform interpolation to obtain the full volume of 155 slices for BRATS and 160 slices for UK Biobank. We evaluate Score-MRI on the same test split as X-Diffusion using standard reconstruction metrics i.e 3D PSNR, and SSIM.
- Adapted TPDM Lee et al. (2023) TPDM allows for sparse input training from two orthogonal views, and subsequently perform fusion of the outputs from the two diffusion models. To adapt TPDM to our experiment on the number of slice input, prior to the fusion module, we condition on n slices from the full volume uniformly sampled in the volume range [1,155] for BRATS and [1,160] for UK Biobank. We evaluate TPDM on the same test split as X-Diffusion using standard reconstruction metrics i.e 3D PSNR, and SSIM.
- Used consistent evaluation metrics across all models, interpreting results with an understanding of input differences.
- Conducted ablation studies to assess the influence of conditioning slices on reconstruction quality.
- Included qualitative analyses and expert evaluations to complement quantitative metrics.

## D.4 SPINE CURVATURE ANALYSIS

For the spine segmentation on UK Biobank, we use a UNet++ model (Zhou et al., 2018) with Dice Loss. We use a model trained to predict curves on DXA on UK Biobank (Bourigault et al., 2022)). We measure the Pearson correlation factor (Bourigault et al., 2022) of spine curvature measured on the generated MRIs where the input is a single MRI coronal slice, a single sagittal slice, or from the paired DXA, against the curvature of reference real MRIs of the same samples. The correlation coefficients are 0.89 for the coronal MRIs, 0.88 for the sagittal MRIs, and 0.87 for the DXAs on the test set of 308 human-annotated angles. We can then bin the curvature, $\kappa$, of the spines under different scoliosis categories based on human-annotated angles: *mild*: $0.06 < \kappa < 0.12$, *moderate*: $0.12 \leq \kappa < 0.15$, and *severe* $\kappa \geq 0.15$. We show the results in Figure XIV. This illustrates that the generated MRIs preserve the spine curvature from normal to severe scoliosis cases.

## D.5 ERROR PLOTS AS DISTANCE FROM INPUT SLICE

We show in Figure XV an error plot of the MSE error as a function of distance from the input slice index 78. We can see that as the distance increases the eror increases , before slowly decreases as the information content is reduced at the boundary and the model can predict this accurately.

## D.6 INPUT SLICE TO FIG.6

We show in FigureXVI the input slice used to generate the 3D volume.

## D.7 LARGE NUMBER OF INPUT SLICES

We show in Figure XVII as the number of input slices increases to create a dense input (120 input out of 155), the baselines outperform X-Diffusion iun predicting the full volume. This highlights the specialty of X-Diffusion for reconstructing sparse inputs of the MRI.

## D.8 ADDITIONAL METRICS FOR EVALUATION

For additional transparency and clarity of our results,we report additional metrics and details to the original ones reported in Table 1. Specifically, we add the standard deviations of the PSNR metric

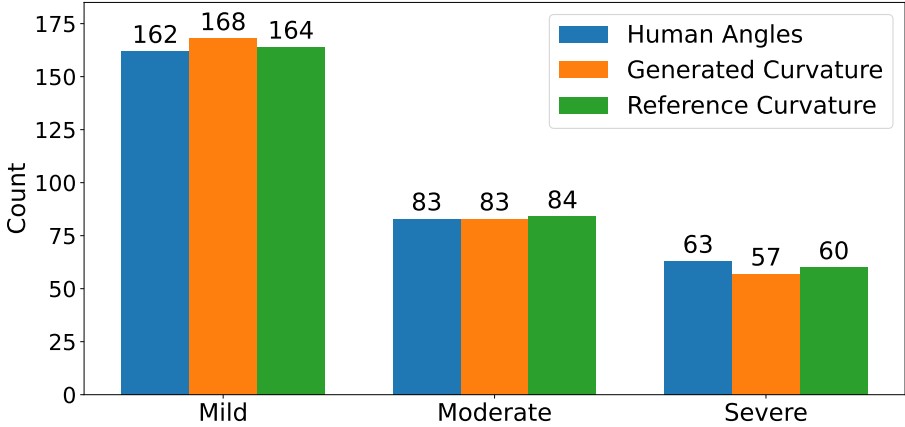

Figure XIV: **Scoliosis Categories of Generated MRIs**. We show spine curvature predicted *v.s.* reference curvature and human annotated angles for scoliosis categories in section 5.2. The barplot indicates that our generated MRIs maintain almost the same distribution of scoliosis categories for then set of 308 patients annotated in the UK Biobank.

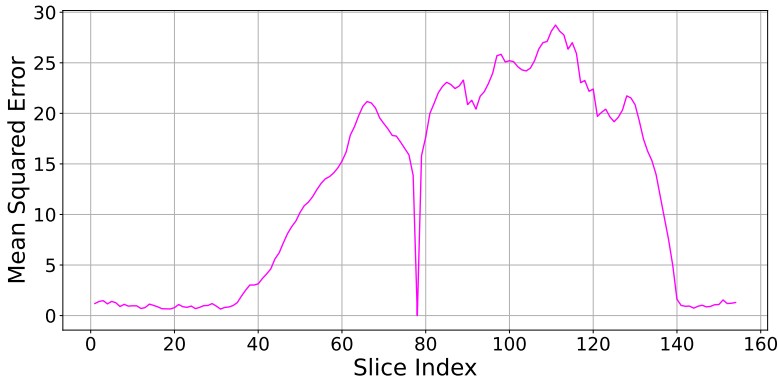

Figure XV: **Error Plots.** We show generated MRI MSE Error *v.s.* output slice index for input slice 78 of samples similar to the ones in figLabel4.

in Table IX. Table X reports the LPIPS and SSIM metrics as well which corresponds to the PSNR metric.

| Test 3D PSNR ↑ STD ↓ | ScoreMRI | | TPDM | | X-Diffusion | |
|---|---|---|---|---|---|---|
| | **BR** | **UK** | **BR** | **UK** | **BR** | **UK** |
| **1 slice** | $9.37 \pm 1.46$ | $8.54 \pm 2.12$ | $10.48 \pm 1.29$ | $9.29 \pm 1.83$ | $\mathbf{23.10 \pm 1.1}$ | $\mathbf{22.42 \pm 1.58}$ |
| **2 slices** | $10.25 \pm 1.09$ | $9.16 \pm 1.45$ | $10.86 \pm 1.22$ | $9.99 \pm 1.78$ | $\mathbf{25.20 \pm 1.0}$ | $\mathbf{23.04 \pm 1.52}$ |
| **3 slices** | $10.68 \pm 1.07$ | $10.42 \pm 1.42$ | $11.33 \pm 1.15$ | $11.09 \pm 1.69$ | $\mathbf{29.43 \pm 0.08}$ | $\mathbf{25.26 \pm 1.40}$ |
| **5 slices** | $12.37 \pm 1.08$ | $11.88 \pm 1.43$ | $14.13 \pm 1.12$ | $12.62 \pm 1.67$ | $\mathbf{31.25 \pm 0.09}$ | $\mathbf{26.85 \pm 1.31}$ |
| **10 slices** | $14.31 \pm 1.06$ | $13.24 \pm 1.41$ | $16.65 \pm 1.07$ | $15.88 \pm 1.59$ | $\mathbf{33.27 \pm 0.08}$ | $\mathbf{27.44 \pm 1.29}$ |
| **31 slices** | $29.24 \pm 1.02$ | $19.01 \pm 1.39$ | $31.48 \pm 0.99$ | $21.70 \pm 1.25$ | $\mathbf{35.48 \pm 0.08}$ | $\mathbf{29.01 \pm 1.24}$ |

Table IX: **Model Performance on Test Brain Data and Whole-Body MRIs (Extension with standard deviation (STD))**. We compare the MRI reconstruction for baselines ScoreMRI (Chung & Ye, 2022), TPDM (Lee et al., 2023), and our X-Diffusion model for varying input slice numbers in training and inference. We report the mean 3D test PSNR on BRATS (**BR**) brain dataset and the UK Biobank body dataset (**UK**). The results showcase huge improvement over the baselines, especially on the small number of input slices (particularly at 1).

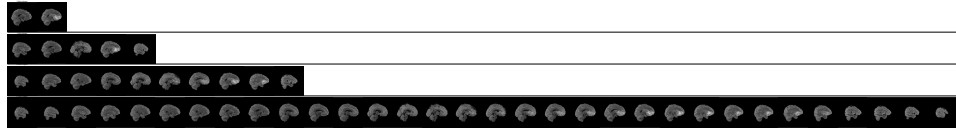

Figure XVI: **Input Slices to Fig6 Main Paper**. We show the input slices used to generate the brain volume shown in Figure 6 comparing baselines and our model X-Diffusion. *1st row* are the 2 inputs for the **2 slices** input model. *2nd row* are the inputs for the **5 slices** input model. *3rd row* are the inputs for the **10 slices** input model. *4th row* are the inputs for the **31 slices** input model.

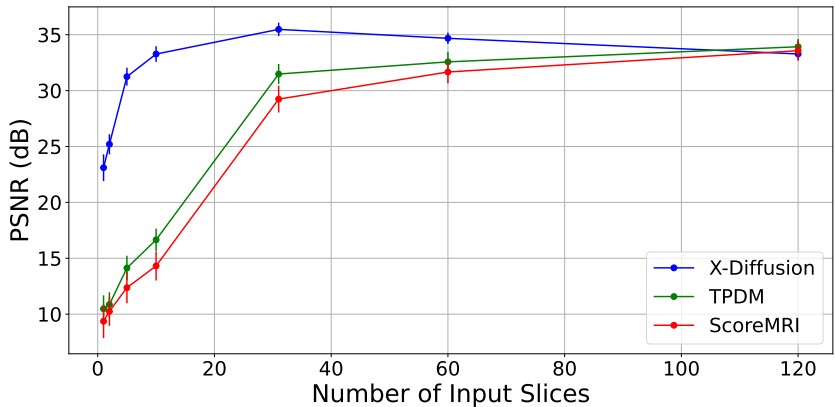

Figure XVII: **Effect of Number Of Slices**. We plot the test PSNR *v.s.* number of input slices for X-Diffusion and our baselines i.e. TPDM (Lee et al., 2023) and ScoreMRI (Chung & Ye, 2022) on the brain MRI dataset using volume averaging. We show the STD of each run to account for randomness.

### D.9 ABLATION STUDY ON TEST TIME AUGMENTATION (TTA)

We perform a series of transformations : horizontal and vertical flips, rotation in degrees [0, 90, 180, 270], and scaling [1,2,4] on the test images and we average them for the final predictions. We apply the augmentations above (flips, rotation, scale) on the test images and we pass these augmented batches through model. We then reverse the transformations for each batch and merge predictions via mean to obtain the output.

We show below in the table XI the summary of the experiments on test time data augmentation for our best model (31 slices). We show little improvement in PSNR () using TTA over the baseline. We also perform downstream segmentation task to measure the effect of TTA on the quality of brain tumour generation compared to baseline X-Diffusion mean dice score. X-Diffusion with TTA achieves a brain tumour segmentation overall dice score of **83.36** compared to **83.09** for the baseline which suggests TTA has improves the quality of brain MRI generation but this effect is limited.

### D.10 TRYING FASTER DIFFUSION MODELS

We experiment fine-tuning from more modern SD diffusion weights than SD 1.O. We previously shown the beneficial effect of large pre-training on objaverse dataset with view-dependent images with significantly better performance fine-tuning Zero-123 with Zero-123 checkpoints. We perform ablation experiments fine-tuning the model from SD 1.0Rombach et al. (2021), SD 2.1Rombach et al. (2022b), SD-XL Podell et al. (2023). We also fine-tune the model with more recent Zero-123-XL checkpoint Liu et al. (2023). We summarize the results in Table XII.

### D.11 FROM ANISOTROPIC TO ISOTROPIC VOLUME GENERATION

We evaluate the capacity of our model to reconstruct SR scans. Scans in BRATs are released re-sampled to isotropic resolution $1mm^3$. We perform the following experiment by downsampling the

| Method | PSNR↑ | SSIM↑ | | | LPIPS ↓ |
|---|---|---|---|---|---|
| | | Axial | Coronal | Sagittal | |
| ScoreMRI | 29.24 | 0.663 | 0.671 | 0.667 | 0.118 |
| TPDM | 31.48 | 0.814 | 0.806 | 0.797 | 0.087 |
| X-Diffusion (ours) | 35.48 | 0.891 | 0.889 | 0.881 | 0.035 |

Table X: **Model Performance on Test Brain Data**. The MRI reconstruction for baselines ScoreMRI (Chung & Ye, 2022), TPDM (Lee et al., 2023), and our X-Diffusion model for 31 input slices numbers in training and inference. We report the mean 3D test PSNR, SSIM, and LPIPS on BRATS (**BR**) brain dataset.

| Method | PSNR↑ | SSIM↑ | | |
|---|---|---|---|---|
| | | Axial | Coronal | Sagittal |
| X-Diffusion (baseline) | 35.48 | 0.891 | 0.889 | 0.881 |
| X-Diffusion + TTA (h/v flips) | 35.60 | 0.894 | 0.892 | 0.884 |
| X-Diffusion + TTA (rotation) | 35.59 | 0.894 | 0.890 | 0.882 |
| X-Diffusion + TTA (scale) | 35.60 | 0.895 | 0.891 | 0.883 |
| X-Diffusion + TTA (all) | 35.61 | 0.896 | 0.893 | 0.884 |

Table XI: **Test Time Augmentation (TTA) Effect on Model Performance on Test Set Brain Data**. We compare the MRI reconstruction for our X-Diffusion model using 31 input slices numbers in training and inference. We report the mean 3D test PSNR and SSIM on BRATS (**BR**) brain dataset with and without TTA. The results suggest slight improvement using TTA over the baseline.

z-dimension to 2mm and aiming to generate isotropic scans 1 x 1 x 1 from anisotropic 1 x 1 x 2. We show the results in Table XIII.

## D.12 ABLATION STUDY ON THE TIME STEPS T DURING INFERENCE

We study the effect of using different time steps $T$ during inference on the test performance of the reconstruction of X-Diffusion. We show the results in Figure XVIII. The performance for both datasets BRATS and UKBB plateau after 800 steps. This indicates we could reduce the number of time down to 800 to improve efficiency.

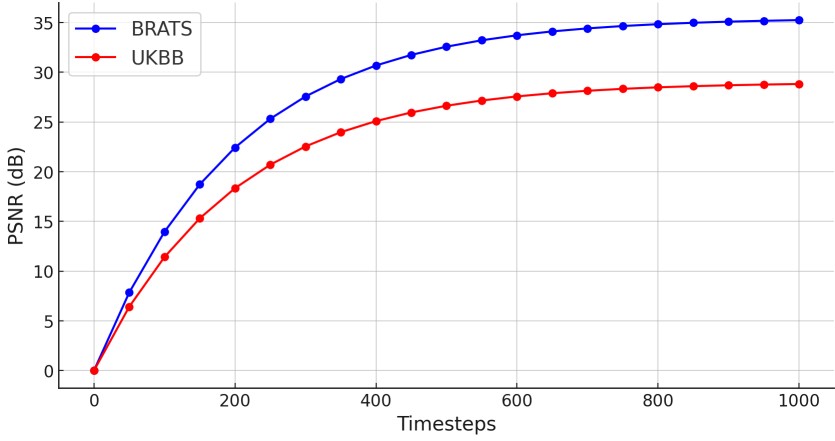

Figure XVIII: **Timesteps Impact on Performance at Inference.** We show the variation of PSNR for varying number of timesteps up to 1000(value used in main paper). The performance for both datasets BRATS and UKBB plateau after 800 steps. We could reduce the number of time steps up to 800 to improve efficiency.

| | Runtime(s) | PSNR↑ | SSIM↑ | | |
|---|---|---|---|---|---|
| **Method** | | | Axial | Coronal | Sagittal |
| X-Diffusion SD 1.0 (baseline) | 141.5 | 27.86 | 0.524 | 0.538 | 0.521 |
| X-Diffusion SD 2.1 | 141.6 | 27.94 | 0.528 | 0.543 | 0.524 |
| X-Diffusion SD XL | 141.9 | 28.13 | 0.586 | 0.597 | 0.583 |
| X-Diffusion Zero-123 (baseline) | 141.5 | 35.48 | 0.891 | 0.889 | 0.881 |
| X-Diffusion Zero-123-XL | 141.5 | 35.71 | 0.896 | 0.890 | 0.883 |

Table XII: **Effect of Fine-Tuning from different pre-trained weights on Model Performance on Test Set Brain Data**. We report the inference runtime in (s), the average 3D PSNR and SSIM on axial, coronal, and sagittal planes. We compare the MRI reconstruction for our X-Diffusion model using 31 input slices numbers in training and inference.

| | PSNR↑ | SSIM↑ | | |
|---|---|---|---|---|
| **Method** | | Axial | Coronal | Sagittal |
| X-Diffusion (isotropic) | 35.48 | 0.891 | 0.889 | 0.881 |
| X-Diffusion (anisotropic) | 33.14 | 0.848 | 0.841 | 0.837 |

Table XIII: **Evaluation of X-Diffusion Performance for Anisotropic to Isotropic Setup**. We compare the MRI reconstruction for our X-Diffusion model trained from multi-view from isotropic (1 x 1 x 1) voxels (baseline) to anistropic (1 x 1 x 2) setting downsampling by factor 2 the z dimension *second row*. We report the mean 3D test PSNR and SSIM on BRATS brain dataset.

### D.13 ASSESS CONFIDENCE IN PREDICTIONS BY CONFIDENCE INTERVALS AND CONFORMAL PREDICTION

To assess confidence in predictions from our diffusion-based model, we use the definition by Horwitz & Hoshen (2022) below. Let a calibration set be defined as $\{x_i, y_i\}_{i=1}^N$ where $x_i, y_i \in [0, 1]^{M \times N}$ are the generated and target image respectively. Our goal is to construct a confidence interval around each pixel of $\hat{y}_i$ such that the true value of the pixel lies within the interval with a probability set by the user. Formally, for each pixel we construct the following interval:

$$\mathcal{T}(x_{imn}) = \left[\hat{l}(x_{imn}), \hat{u}(x_{imn})\right] \tag{9}$$

where $\hat{l}, \hat{u}$ are the interval lower and upper bounds. To provide the interval with statistical soundness, the user selects a risk level $\alpha \in (0, 1)$ and an error level $\delta \in (0, 1)$. We then construct intervals such that at least $1 - \alpha$ of the ground truth pixel values are contained in it with probability of at least $1 - \delta$. That is, with probability of at least $1 - \delta$,

$$\mathbb{E}\left[\frac{1}{MN}\left|\{(m, n) : y_{(m,n)} \in \mathcal{T}(x)_{(m,n)}\}\right|\right] \geq 1 - \alpha, \tag{10}$$

where $x, y$ are a test sample and label originating from the same distribution as the calibration set.

Setting $\delta$ to 0.05, we are confident at the 5% level that the true value in Table last row for X-Diffusion lies in [34.49, 35.55] for model trained on BRATS (BR) and [27.77,30.25] for UK Biobank (UK) model. We also compute the 99% confidence interval as comparison. We are confident at the 1% level that the true value in Table last row for X-Diffusion lies in [34.11, 35.72] for model trained on BRATS (BR) and [27.29,30.78] for UK Biobank (UK) model.

### D.14 ANALYSIS ON WHITE MATTER AND CORTICAL VOLUME OF BRAINS

In order to study cortical volumes, we use a brain *parcellation* module from Li et al. (2017). It works by splitting the brain in 160 different structures via CNNs, in a similar manner as geodesic information flows Cardoso et al. (2012). The volumes for each structure is computed via binary label map representation using the software 3D Slicer, Segment Statistics Module. We compute 2-sample t-tests for each structure on the test set and report the test statistics and p-values. We compare generated and real periventricular white matter on the test set by additioning left and right volume mean. The mean difference is of $194.85 \pm 83 \ mm^3$, p-value > 0.05. We compare overall

white matter volume from our parcellation, between generated 28533.05 and real 28802.42, mean difference of $269.37 \pm 105 \ mm^3$, p-value>0.05. Both p-values are not significant at 5% level which suggest generated brains preserve white matter volume.

### D.15 THE IMPORTANCE OF THE VIEW USED IN PROCESSING THE MRI

In the UK Biobank, the data is collected based on the transversal (axial) plane. This might suggests that processing the UKBB MRIs from that axis would yield better results. Table XIV compares the different planes in generating MRIs using our X-Diffusion, or by combining multiple planes using our volume averaging technique proposed in Section 3.2. It clearly shows that there is no significant difference between the different planes, but using the multi-view volume aggregation indeed yields improved performance. This is explained by the fact that deep learning models like X-Diffusion benefit from increase and variety of the size of training data to generalize, which benefit from exposure to as many views as possible.

| Method | PSNR↑ | SSIM↑ | | |
| --- | --- | --- | --- | --- |
| | | Axial | Coronal | Sagittal |
| X-Diffusion (axial) | 34.91 | 0.859 | 0.858 | 0.854 |
| X-Diffusion (coronal) | 35.17 | 0.862 | 0.860 | 0.857 |
| X-Diffusion (sagittal) | 34.23 | 0.847 | 0.844 | 0.841 |
| X-Diffusion (multi-view) | 35.48 | 0.891 | 0.889 | 0.881 |

Table XIV: **Comparison on Conventional Planes and Multi-View on Test Set Brain Data**. We compare the MRI reconstruction for our X-Diffusion model trained from input axial, coronal, or sagittal. We report the mean 3D test PSNR and SSIM on BRATS (**BR**) brain dataset.

### D.16 BLAND-ALTMAN PLOT FOR SPINE CURVATURE

We show the Bland-Altman plot for the curvature of spines of the generated MRIs with X-Diffusion *v.s.* the curvature of the Ground Truth MRIs in Figure XIX.

### D.17 TEST-TIME OPTIMISATION (TTO)

TTO has proved performant in improving the performance of diffusion-based models by encouraging diversity in model output and ensuring that the generated data is not overly deterministic (Shi et al. (2023); Pu et al. (2023); Sargent et al. (2024)). A standard way of applying TTO is through entropy minimisation of the logits. This is is achieved by dynamically adjusting the noise predictions during the iterative denoising process.

This is how we proceed. From the initialised latent variable (noisy image), we run the pre-trained diffusion model. We obtain predicted noise and logits and we compute the entropy loss of the logits. Lower entropy suggests higher confidence. The goal is to maximize entropy in the loss function to increase diversity. Then we adjust the predicted noise with entropy optimization with an entropy weight factor that we set to 0.01. We adjust the classifier-free guidance and update $z_t$ with the adjusted noise before converting this latent sample into an image using the diffusion model decoder.

We observe minor improvement in performance metrics with TTO (see Table Table XV). It is worth noting that TTO relies heavily on the original accuracy of the predictor. Gradient computation requires backpropagation through the logits, which introduces major computational load. Computational cost with TTO compared to inference without TTO is increased by three times. We report the improvement in terms of PSNR, SSIM, LPIPS metrics and the inference cost in Table XV.

| Method | Runtime(s) | PSNR↑ | SSIM↑ | | | LPIPS ↓ |
| --- | --- | --- | --- | --- | --- | --- |
| | | | Axial | Coronal | Sagittal | |
| X-Diffusion | 141.461 | 35.48 | 0.891 | 0.889 | 0.881 | 0.035 |
| X-Diffusion + TTO | 424.383 | 35.97 | 0.894 | 0.893 | 0.883 | 0.028 |

Table XV: **Model Performance on Test Brain Data and Whole-Body MRIs**. We compare the MRI reconstruction performance and inference cost for our X-Diffusion model with and without test-time optimisation (TTO) for 31 input slices. We report the mean 3D test PSNR, SSIM, and LPIPS on BRATS (**BR**) brain dataset.

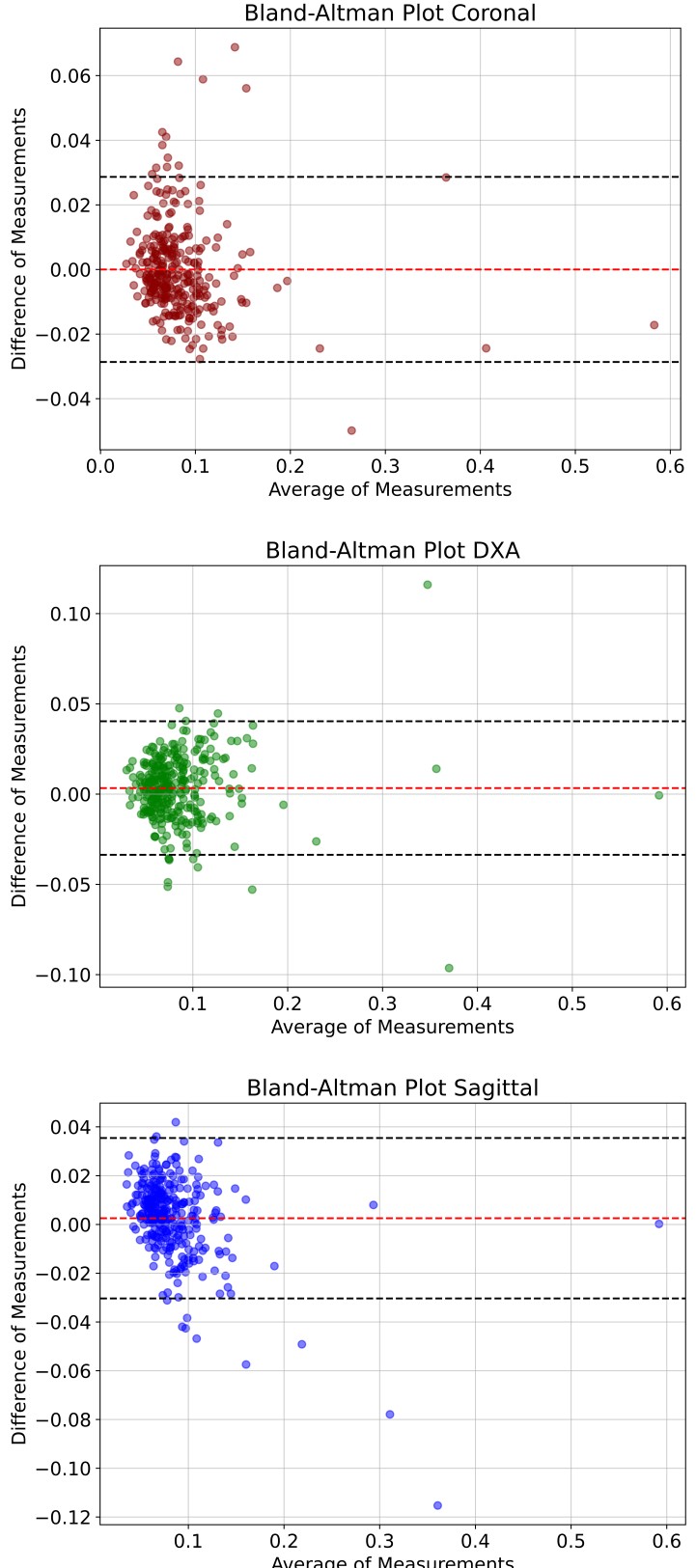

Figure XIX: **Spine Curvature Bland-Altman Plots**. We show for each plot the average of each predicted and ground-truth value on the x-axis and the difference between the predicted and ground-truth value on the y-axis.

