# OpenReview forum: "X-Diffusion: Generating Detailed 3D MRI Volumes From a Single Image Using Cross-Sectional Diffusion Models"
_ICLR.cc/2025/Conference — Submitted to ICLR 2025_

### Official Review · Reviewer_YwSq · 2024-10-25

**Soundness:** 2
**Presentation:** 2
**Contribution:** 2
**Rating:** 5
**Confidence:** 5

**Summary:**

The proposed manuscript explores the use of a conditional generative model which can produce a full 3D MRI volume conditioned only on 1 or MRI slices. The authors conduct experiments with three MRI datasets that span brain, whole-body, and out-of-distribution knee MRI scans. Overall, this study could be improved with a justification of why such an approach that has no test-time data consistency should be favored over conventional deep learning based variational reconstruction methods that are very common and quite robust.

**Strengths:**

*	Multiple datasets are used for the evaluation
*	The diffusion based benchmarks are chosen well and represent some of the latest approaches in the field.

**Weaknesses:**

*	Many of the approaches that are used for MRI reconstruction rely on the notion of data consistency which can ensure that there is consistency between the reconstructed imaging data and the extent of signals that are actually obtained. The proposed approach provides no sense of such confirmation.
*	It is mostly an impractical scenario where a single high-resolution slice will be obtained instead of obtaining multiple undersampled slices. So the overall premise of the work unfortunately seems more academic rather than something that would have a clinical impact.
*	In prior work, there is a very incomplete analysis of work in image reconstruction and super-resolution. It is unclear what “classical” means. based on the references, does classical mean anything without foundation models? This doesn’t seem accurate.
*	In figure 4, the acquisition is actually axial and study depicts the acquisition to be coronal. Since there is always a fully sampled dimension readout dimension in MRI, the reformatting of this 3D volume does not accurately convey the representational capacity of the model. Actual slice traversal needs to be in the *true* slice direction.
*	A better evaluation would be show a progression of PSNR values the further away a test slice is being computed from a conditioning slice.
*	All datasets had 155 slices. In the case of using 31 conditioning slices, it is not clear why one should use this approach instead of just undersampling the 3D data by 5x and then reconstructing with variational methods. These methods can easily provide accelerations of 8-10x with much higher metrics.
*	In figure 6, up until 10-13 slices, there are significant hallucinations in the structure and contrast of the grey and white matter. These results would cause radiologists quite a bit of anxiety! Like the previous point, at this point, why not just use a variational reconstruction method?
*	The results in Figure 6 and the massive impact in hallucinated details entirely reduces confidence in all 1-5 slice results. This further shows how poor a metric PSNR is.
*	Total brain volume is not the correct metric to assess fidelity. Cortical volumes and folding patters would be better. One would hypothesize that it is likely trivial to regress total brain volume from a single brain MRI slice anyway.
*	It is pretty shocking to me why the model should be able to work in the OOD setting to synthesize knee MRI scans? Can the authors please provide a rationale why this should work in the first place? The UKB dataset includes IDEAL fat-water sequences that are MUCH lower resolution than the 2D TSE knee MRI fastMRI sequences that have a proton density contrast. Why should the model know this distribution Section 6.3 attempts to answer this but simply claiming Stable Diffusion training and large non-domain specific datasets unfortunately does not seem convincing.

**Questions:**

*	Since H = W = D in this study, what would happen when the data is non-isotropic, as is most common in MRI applications and protocols.
*	For the example provided in Figure 5, it is interesting to think about why the method should work in the first place? Given that this is an ill-posed problem, why should a model be able to predict a slice that is 20 slices away accurately?
*	The better test would be to pick a conditioning slice that does not have a tumor but from a patient with a tumor. Then, will the model correctly produce a tumor? That is the real test. I hypothesize that there is no reason a model should be able to predict that. A brief sentence or two is included in the paper but this analysis can be made more rigorous.
*	Spine curvature: What is the distribution of spine curvature? One may think this is regression to the mean of healthy spines. showing a Bland-Altmann or regression pot where would be better.

---

> ### Author Response · Authors · 2024-11-21
> **Response to Reviewer's Comments and Questions**
>
> We thank the reviewer for their insightful comments and relevant questions, which we tried to address to the best of our ability. We updated the paper PDF with section D on Additional Discussions in supplementary material for the rebuttal.
>
> >_Many of the approaches that are used for MRI reconstruction rely on the notion of data consistency which can ensure that there is consistency between the reconstructed imaging data and the extent of signals that are actually obtained. The proposed approach provides no sense of such confirmation._
>
> While our X-Diffusion is indeed a feed-forward reconstruction method and not an optimization-based method ( improved speed advantage), it does incorporate multi-view consistency since it generates the MRI volume from independent different views, each having a novel perspective and output and the final output is the average volume of these independent volumes. However, we agree with the reviewer that optimization-based methods have an edge in terms of incorporating a feedback loop to improve the output. We run an experiment to do Test Time Augmentation to improve the output of X-Diffusion based on feedback from the output and input signals and report the results in Table XI and Sec D.9 in the updated Appendix.
>
> >_It is mostly an impractical scenario where a single high-resolution slice will be obtained instead of obtaining multiple undersampled slices. So the overall premise of the work unfortunately seems more academic rather than something that would have a clinical impact._
>
> We kindly agree with the reviewer that these generated “pseudo MRISs” are not a subsite for actual MRI scans as we described briefly on page 2 “It is important to note that the generated MRIs are not clinical replacements for true MRIs yet, but could provide a quick, affordable, and informative “pseudo-MRI" before conducting a full MRI examination.”. So we envision in the future this can be used in an active setup, while the patient is conducting a scan the model reconstructs the MRI and informs the technician where to scan next, reducing reducing in the scanning process, and hence reducing the time in conducting the MRI scan, reducing the cost of running MRI scans, and potentially saving countless lives. We hope that our X-Diffusion (especially if published at ICLR 2025) will act as the first step towards this future we envision. We include Sec. D1, and D.2  in the updated Appendix to highlight this discussion.
>
> >_In prior work, there is a very incomplete analysis of work in image reconstruction and super-resolution. It is unclear what “classical” means based on the references, does classical mean anything without foundation models? This doesn’t seem accurate._
>
> We apologize for the minor issue in the related works section. We have modified the related works section to be more comprehensive and include the pre-learning works and is now integrated into the paper PDF.
>
> >_In figure 4, the acquisition is actually axial and the study depicts the acquisition to be coronal. Since there is always a fully sampled dimension readout dimension in MRI, the reformatting of this 3D volume does not accurately convey the representational capacity of the model. Actual slice traversal needs to be in the true slice direction._
>
> While it's possible to limit the slice traversal to only the true slice collection plane, there is no specific reason why other directions will not be helpful given the model is a learning-based model and would benefit greatly from scaling, diversity, and augmentation. In fact, the multi-view slicing is “of particular interest to the medical imaging community and constitutes - to the best of my knowledge - a unique contribution of the paper” as reviewer F5BK finds it. We run an ablation study to clarify this particular point about the original view vs other views effect in UK Bioban, which shows there is no significant difference between the different planes, but using the multi-view volume aggregation indeed yields improved performance. This is explained by the fact that deep learning models like X-Diffusion benefit from the increase and variety of the size of training data to generalize, which benefits from exposure to as many views as possible. We show the results in the appendix updated PDF in sec D.15 in Table XIV.
>
> >_A better evaluation would be to show a progression of PSNR values the further away a test slice is being computed from a conditioning slice._
>
> We agree with the reviewer on this point and we show error plots as a function of distance from the conditioning slice in updated Appendix Sec D.8 Fig XV. We can see that as the distance increases the error increases before slowly decreases as the information content is reduced at the boundary and the model can predict this accurately.

---

> ### Author Response · Authors · 2024-11-22
> **Following Response to Reviewer's Comments and Questions**
>
> >_All datasets had 155 slices. In the case of using 31 conditioning slices, it is not clear why one should use this approach instead of just undersampling the 3D data by 5x and then reconstructing it with variational methods. These methods can easily provide accelerations of 8-10x with much higher metrics._
>
> This is exactly true, the baselines can do this (easy) job well, and actually, by looking at Table 1 we can see the difference between the baselines and our X-Diffusion shrinks to almost 0 and actually in plot fig X in the updated appendix, if input slices go 100, the baselines beats our X-Diffusion. That is why the main focus of our work, different from all previous works in MRI reconstruction is that we reconstruct accurate MRIS from extremely sparse input, as clearly explained in the introduction and Fig 1 of the paper. This is a new area of research not typical in MRI reconstruction, and we hope our X-Diffusion paper ( if accepted to ICLR 2025 for example ) will open this door for other works in the field.
>
> >_In Figure 6, up until 10-13 slices, there are significant hallucinations in the structure and contrast of the grey and white matter. These results would cause radiologists quite a bit of anxiety! Like the previous point, at this point, why not just use a variational reconstruction method?_
>
> While the variation reconstruction method will work for a large number of slices like 31, or more, it fails miserably with very few slices as can be seen in the same Fig6 with 2-slice input for the baselines in the top two rows.
>
> >_The results in Figure 6 and the massive impact in hallucinated details entirely reduce confidence in all 1-5 slice results. This further shows how poor a metric PSNR is._
>
> While PSNR is not a perfect metric, we include SSIM like in Table II in the appendix. Additionally, we include LPIPS metrics for the main results of the paper and add them in the updated appendix PDF in Section D.10 Table X.
>
> >_Total brain volume is not the correct metric to assess fidelity. Cortical volumes and folding patterns would be better. One would hypothesize that it is likely trivial to regress total brain volume from a single brain MRI slice anyway._
>
> We perform the requested cortical volume measurements and folding patterns and report the results in the updated Appendix PDF Section D.14.
>
> >_It is pretty shocking to me why the model should be able to work in the OOD setting to synthesize knee MRI scans. Can the authors please provide a rationale for why this should work in the first place? The UKB dataset includes IDEAL fat-water sequences that are MUCH lower resolution than the 2D TSE knee MRI fast MRI sequences that have a proton density contrast. Why should the model know this distribution Section 6.3 attempts to answer this but simply claiming Stable Diffusion training and large non-domain specific datasets unfortunately does not seem convincing._
>
> We are not the first to show the OOD capability of diffusion models. Score-MRI showed that their model trained solely on proton density fat suppression (PDFS) coronal knee scans achieves remarkable performance in generating brain MRI slices (axial plane) (see Fig 9 of Score-MRI paper (https://arxiv.org/pdf/2110.05243).) The idea advanced in Score-MRI comes from the posterior sampling in the diffusion being very robust to distribution shifts. Moreover, we compared the PSNR of two randomly sampled knee and brain MRIs and obtained an average PSNR of 18.34 over 30 sampling pairs. This is a relatively high PSNR value which suggests that the knee and brain MRI share to some extent patterns in their features.
>
> >_Since H = W = D in this study, what would happen when the data is non-isotropic, as is most common in MRI applications and protocols._
>
> Zero padding and resizing cropping are all common and naive approaches to handling such scenarios. Actually, in our experiments with UKBB, we pad with zeros all dimensions to make them equal.

---

> ### Author Response · Authors · 2024-11-22
> **Follow-up Response to Reviewer's Comments Questions**
>
> >_For the example provided in Figure 5, it is interesting to think about why the method should work in the first place. Given that this is an ill-posed problem, why should a model be able to predict a slice that is 20 slices away accurately?_
>
> This can be answered easily as follows. X-Diffusion is pre-trained with 5 billion images in LAION and fine-tuned on 48 K MRI samples with almost 17.9M 2D images. This gives the model the ability to generalize to samples that are especially similar to the domain in which it was trained. For example, Zero123 paper showed how finetuning Stable Diffusion on 3D object models, allows the diffusion model to generate 3D objects from a SINGLE IMAGE, even though it is an ILL-POSED problem, as there might be many ways the images can be formed. However, people in the 3D computer vision community have verified these approaches on 3D datasets, and actually, the results are very convincing and even 3D geometrically accurate [a][b]. The same rationale is applied here with X-Diffusion generating MRI 3D volumes from a single/few slices, and that is actually the original motivation for why this whole project came into play.
>
> [a] Magic123: One Image to High-Quality 3D Object Generation Using Both 2D and 3D Diffusion Priors, (ICLR 2024)
>
> [b] LGM: Large Multi-View Gaussian Model for High-Resolution 3D Content Creation, (ECCV 2024)
>
>
> >_The better test would be to pick a conditioning slice that does not have a tumor but from a patient with a tumor. Then, will the model correctly produce a tumor? That is the real test. I hypothesize that there is no reason a model should be able to predict that. A brief sentence or two is included in the paper but this analysis can be made more rigorous._
>
> We indeed performed this experiment in the original paper appendix, it is in appendix Sec B,3 and Figure VI and Table I. We on purpose picked the conditional slice to not include the tumor and report the results as input gets closer to the tumor. The study briefly shows that as the input conditioning slice gets closer to the center of the tumor, its prediction of the entire tumor increases. However, even with no intersection, the performance is still decent.
>
> >_Spine curvature: What is the distribution of spine curvature? One may think this is a regression to the mean of healthy spines. showing a Bland-Altmann or regression pot where would be better._
>
> A healthy spine in coronal view would be straight, If not healthy there would be a curve in the spine, and curvature we can measure. We apologize for not being clear, we provide Pearson’s correlation, which is a measure of how well the curve matches. We show the requested additional plots in the updated appendix sec D.16 Fig XIX (Bland-Altman)  and sec D.8 Fig XIV (scoliosis severity categories). We can see that the curvature creates varied distributions of scoliosis severity and the X-Diffusion generations preserve these categories of scoliosis severity.

---

> > ### Author Response · Authors · 2024-11-25
> > **Follow-up**
> >
> > Dear reviewer,
> >
> > We would like to know if we adressed your comments. Please let us know whether we need to perform any additional experiments.

---

> ### Comment · Reviewer_YwSq · 2024-11-26
> **Feedback to Responses**
>
> > The lack of data consistency of proposed approach.
>
> The authors have added test time augmentation as well as multi-view consistency. However, all of these are data-driven metrics and do not necessarily enforce data consistency with the acquired raw data. Note that the optimization-based approaches mentioned here are actually commonly unrolled variational models, which are more efficient than diffusion models.
>
> > Pseudo-MRIs:
>
> Localizer/scout scans are always performed first and provide this ability anyway, so it is not clear what the additional overhead of the proposed method adds.
>
> > Hallucinations when <13 slices are used for conditioning:
>
> As shown in the figures in the manuscript, there are extensive hallucinations in the generated images. Independent of the metrics, which are low to begin with, these hallucinations would not be high-fidelity enough for any clinical application.
>
> Thank you for adding the new analysis regarding cortical volumes and the spinal curvature. These results make the paper stronger.
>
> > Figure 5 Discussion:
>
> Thank you for sharing the insights from the natural 3D imaging community. However, there are some differences because in a medical image the image that is being reconstructed is dense. In contrast, for general-purpose 3D imaging, we are simply reconstructing the outer surfaces of images which are sparser. In any case, it is not clear that there is any theoretical explanation possible at this particular point, but the comment was mostly made to get the author's perspectives.
>
> Overall, from a methodological perspective, there are some nice new innovations that this work provides. However, the lack of applicability to clinical settings, as mentioned by the authors, ultimately limits the impact that the proposed method could add. I will raise my score from a 3 to a 5.

---

> ### Author Response · Authors · 2024-11-27
> **followup on feedback**
>
> We thank the reviewer for increasing the score to 5. Here is the response to the additional concerns raised during the discussions.
>
> >_The authors have added test time augmentation as well as multi-view consistency. However, all of these are data-driven metrics and do not necessarily enforce data consistency with the acquired raw data. Note that the optimization-based approaches mentioned here are actually commonly unrolled variational models, which are more efficient than diffusion models._
>
> We added in Section D.17 Table XV an actual optimization-based approach of the X-Diffusion model for each acquired raw data during test time. The results show a slight improvement in the quality of the reconstructed MRI  compared to the feed-forward only method (+0.5 PSNR), but with an additional 3X time required during inference. This explains why we resort to a feed-forward diffusion instead of an optimization approach. variational models indeed are more efficient than Diffusion, but they typically struggle with detailed output generalization (e.g. VNNs [a] compared to Stable Diffusion )
>
> [a] A Variational neural network for image restoration based on coupled regularizers, Multimedia Tools and Applications, 2023
>
> >_Localizer/scout scans are always performed first and provide this ability anyway, so it is not clear what the additional overhead of the proposed method adds._
>
> In MRI, the process of acquiring images involves filling k-space, a matrix that stores raw data before image reconstruction. Traditional MRI scans often employ Cartesian trajectories, where k-space is filled line by line. This method can be time-consuming, as each line requires a separate phase-encoding step, contributing to longer scan durations [a]. Localizer or scout scans are preliminary images used to determine the position and orientation of subsequent imaging slices or volumes. These scans are typically acquired quickly, often within a few seconds, and provide a reference for planning the imaging sequence. However, they usually offer low detail and may cover only a small volume, limiting their diagnostic utility [b]. In contrast, our X-Diffusion by requiring one/few fully sampled slices, reduces the time requirement of covering the original volume and allows us to reach the precision very close to obtaining the complete MRI scan. The whole paper is dedicated to verifying the precision of these generated MRIs (from brain tumour to spine scoliosis). Unlike the current scout MRIs that contain low details or cover small volumes [b], X-Diffusion “pseudo-MRIs” cover large volumes and are detailed as shown in the paper.
>
> [a] Rinck PA. Magnetic Resonance in Medicine. The Basic Text­book of the European Magnetic Resonance Forum., 2024.
>
> [b] https://mrimaster.com/mri-localizer-scout-or-survey-scan/
>
> >_As shown in the figures in the manuscript, there are extensive hallucinations in the generated images. Independent of the metrics, which are low to begin with, these hallucinations would not be high-fidelity enough for any clinical application._
>
> We agree that only PSNR/SSIM are not enough to quantify the usefulness of the generated MRIs. This is why almost half of the paper is about validating the generated MRIs. Figure 5 in the main paper and Table I in the Appendix show the brain tmour detection in the generated MRIs almost matches that of the original MRIs, even though this tmour information was never explicitly used in training X-Diffusion on BRATS. When testing this BRATS-trained model on the healthy brain MRIs of  IXI dataset the predicted tumor is in 9.9% of the real healthy brains and in 11.3% of the generated brain MRIs (sec 6.4), which means the X-Diffusion model is mostly not hallucinating tumors when there is no tumor in the sample. On the other hand for the full body MRIs the spine metrics (Figure XIV and Figure XIX) and the body fat metrics ( figure IX) all further validate these meta information about the generated MRIs, without ever being used in training X-Diffusion model.
>
> >_Thank you for adding the new analysis regarding cortical volumes and the spinal curvature. These results make the paper stronger._
>
> We appreciate the recognition by the reviewer.

---

> ### Author Response · Authors · 2024-11-27
> **followup on feedback 2**
>
> >_Thank you for sharing the insights from the natural 3D imaging community. However, there are some differences because in a medical image the image that is being reconstructed is dense. In contrast, for general-purpose 3D imaging, we are simply reconstructing the outer surfaces of images which are sparser. In any case, it is not clear that there is any theoretical explanation possible at this particular point, but the comment was mostly made to get the author's perspectives._
>
> Yes, indeed there is no theoretical evidence of why such diffusion models generalize well on dense predictions ( like the recent SORA video diffusion models generating completely novel sequences[a] ) or why LLMs trained on next token prediction can solve puzzles and Math Olympic questions and even achieve an IQ higher than the human average [b]. However, LLMs and Diffusion models are being used all over the world currently to create media, write programs, advance Science, and enhance human productivity in all domains of life. We believe our X-Diffusion is a good baby step in this direction of Generative AI helping the struggling costly healthcare sector worldwide.
>
> [a] https://openai.com/index/sora
>
> [b] https://klu.ai/llm-leaderboard
>
>
> >_Overall, from a methodological perspective, there are some nice new innovations that this work provides. However, the lack of applicability to clinical settings, as mentioned by the authors, ultimately limits the impact that the proposed method could add. I will raise my score from a 3 to a 5._
>
> We thank the reviewer for the supporting comments. While we acknowledged potential clinical future work that can be done to verify X-Doffusion feasibility on clinical setup on actual patients,  we believe that would be suitable for a medical journal clinical study and is out of the scope of the current submission to ICLR conference which is about the method novelty, technical execution, and reconstructions results, where the reviewer agrees on (also reviewer `fhnz` just mentioned that). Hence we don’t see this clinical future work as a reason to lower the score of our paper which could result in a rejection. Hence we kindly ask the reviewer if all of their other concerns were addressed properly and if they believe the paper is worthy of acceptance to reconsider raising the score to at least 6 which is still borderline acceptance but at least lowers the risk of rejecting the paper solely based on the clinical feasibility aspect of the paper.

---

### Official Review · Reviewer_fhnz · 2024-10-30

**Soundness:** 2
**Presentation:** 1
**Contribution:** 2
**Rating:** 5
**Confidence:** 4

**Summary:**

The authors propose a diffusion-based approach to reconstruct 3D MRI volumes from a single or multiple 2D slices. Their approach is conditioned by the target rotation and slice index and relies on heavy pre-training. It is applied to brain MRI (with and without tumours), whole-body MRI and knee MRI.

**Strengths:**

- Both the approach and the application appear quite novel.
- Many experiments were performed.

**Weaknesses:**

- The experiments are confusing. It is as if the authors tried all they could think of and let the readers do what they can with the results.
- A major point that is often unclear in the experiments/results is what slice(s) was/were used as input. If the authors show in Figure VI that is has a impact on the results, it is not specified in other sections, e.g. 5.1, 5.2, 5.3, 5.4.
- The way the authors split the BraTS dataset is not clear. On the one hand, they state that the dataset includes 5,880 MRI scans from 1,470 patients (roughly four MRI sequences per patient) and that they split it into Train (n=4704), Validation (n=588), and Test (n=588) sets, without mentioning if it is done at the subject level or not, which could mean data leakage. On the other hand, they say that they just use the FLAIR sequence, so then the numbers do not add up. Please clarify.
- When difference maps are displayed there is no scale, on top of the colormap poorly chosen, so they are impossible to analyse.
- No estimate of variation is provided in the tables.
- The evaluation by medical experts, though valuable, confirms that images generated are realistic but not that they show what they are supposed to, i.e. an image could be realistic but not correspond to the reality of a patient.
- More philosophically speaking, I fail to understand how the authors see what they propose used in practice, which is what they state they aim for. In their scenario, patients would come to the imaging centre, have 2D slices acquired and pseudo 3D volumes reconstructed and then what? Would they go home and maybe be called to come back and have a normal exam later? If so in what case? This does not seem cost-efficient nor practical.
- The related work section should cover super-resolution approaches. Also the Full-Body MRI Analysis paragraph is very generic and pretty empty. See for instance Tunariu et al., British Journal of Radiology, 2020 on the use of whole-body MRI in clinical practice and Küstner et al., Radiology: Artificial Intelligence, 2020 for the automatic analysis methods related to these applications.
- Some references are too generic or do not correspond to the statement they are supposed to comfort, e.g. Tran et al., 2015; Sohl-Dickstein et al., 2015 or Kawar et al., 2022 in the second paragraph of the introduction.

**Questions:**

- Please see weaknesses above.
- Please fix the hyperref links, many references in the Appendix do not point to the good figure, this makes the paper even more difficult to follow.

---

> ### Author Response · Authors · 2024-11-21
> **Response to Reviewer's Comments and Questions**
>
> We thank the reviewer for their insightful comments and relevant questions, which we tried to address to the best of our ability. We updated the paper PDF with section D on Additional Discussions in supplementary material for the rebuttal.
>
>
> >_The experiments are confusing. It is as if the authors tried all they could think of and let the readers do what they can with the results._
>
> We kindly disagree with the statement as other reviewers ( reviewer Haqk and reviewer F5BK) find the paper “ is written in a very clear, concise, and comprehensive way.” and “ problem, motivation, and contributions are clearly stated”. However, we took into account your comment and we added a new subsection in the updated Appendix Section D.3 that explains the evaluation protocol further.
>
> >_A major point that is often unclear in the experiments/results is what slice(s) was/were used as input. If the authors show in Figure VI that it has an impact on the results, it is not specified in other sections, e.g. 5.1, 5.2, 5.3, 5.4._
>
> The test slices are randomly sampled from the entire datasets and fixed throughout the experiments for all the methods including our X-Diffusion and the baselines shown in Table I, etc. as explained in section 4.3 of the paper. We add a detailed subsection about the evaluation slice choice in sec D3 in the appendix
>
> >_The way the authors split the BraTS dataset is not clear. On the one hand, they state that the dataset includes 5,880 MRI scans from 1,470 patients (roughly four MRI sequences per patient) and that they split it into Train (n=4704), Validation (n=588), and Test (n=588) sets, without mentioning if it is done at the subject level or not, which could mean data leakage. On the other hand, they say that they just use the FLAIR sequence, so then the numbers do not add up. Please clarify._
>
> We used the standard test split of BRATS evaluation following previous works. The statement about the 5,880 MRI scans is a description of the dataset, not our training samples, we use the FLAIR sequence. We apologize for any confusion caused by this.
>
> >_When difference maps are displayed there is no scale, on top of the colormap poorly chosen, so they are impossible to analyze._
>
> We added the bar color maps to Figure 4  and updated the PDF.
>
> >_No estimate of variation is provided in the tables._
>
> In Figure 7 we showed the standard deviation on the plot which depicts the same results in Table 1 (very few variations anyway). However, as requested, we added STD on the main Table I, and added it in the Appendix as Table IX in Section D.8.
>
> >_The evaluation by medical experts, though valuable, confirms that images generated are realistic but not that they show what they are supposed to, i.e. an image could be realistic but not correspond to the reality of a patient._
>
> We agree with the reviewer that a proper clinical study that actively relates the accuracy of the generated MRIs with the actual clinical diagnosis of the patients would be ideal. However, we believe it is outside of the scope of this work and can be indeed a perfect further work. We add Sec D.2 in the Appendix to detail possible clinical study procedures to be followed to evaluate the clinical validity of X-diffusion. We intend to do the clinical medical study soon. Please see below the summary of the study objectives :
>
> __Objective:__
> To see if recent generative AI technologies for MRIs (X-Diffusion) are relevant to knee diagnosis. The goal is to validate our pipeline for reconstructing full MRIs from one/few slices with high precision and evaluate its usefulness for clinical assessment of knees. Given expert grading of how abnormal a knee is, we will compare the score given for degeneration of knees between the two sets of knees with and without AI-generated knees.
>
> __Specific Aim.__
> Grade how abnormal a knee is on an external set of 50 reconstructed MRIs from single slices or two slices each using X-Diffusion. Then, compare the score for knee degeneration between the annotated original set of 50 samples from humans and see how they are correlated.
> First, experts will grade degenerative knees on MRI. Then we test X-Diffusion on the same sampled grade. Given any single slice(s) from a degenerate knee, we generate synthetic knees. Finally, we compare the score given as grading for the degeneration of knees between the two sets of knees with and without AI-generated knees.
> Hypothesis.
> The trained X-Diffusion model for generating MRIs is capable of reconstructing unseen MRIs from one/a few slices in the clinic with high precision. The generated MRIs also maintain the diagnosis differentiability that makes them useful for physicians in the clinical setup.

---

> ### Author Response · Authors · 2024-11-21
> **Following Response to Reviewer's Comments and Questions**
>
> >_More philosophically speaking, I fail to understand how the authors see what they propose used in practice, which is what they state they aim for. In their scenario, patients would come to the imaging center, have 2D slices acquired and pseudo-3D volumes reconstructed, and then what? Would they go home and maybe be called to come back and have a normal exam later? If so in what case? This does not seem cost-efficient nor practical._
>
> Currently, our generated “pseudo MRIs” are not a substitute for actual MRI scans as we described briefly on page 2 “It is important to note that the generated MRIs are not clinical replacements for true MRIs yet, but could provide a quick, affordable, and informative “pseudo-MRI" before conducting a full MRI examination.” So we envision in the future this can be used in an active setup, while the patient is conducting a scan and the model reconstructs the MRI and informs the technician where to scan next, reducing redundancy in the scanning process, and hence reducing the time in conducting the MRI scan, reducing the cost of running MRI scans, and potentially saving countless lives. Moreover, in case of corrupted scans which regularly happen in practice, our model could act to generate pseudo-MRIs in the scenario if only one or few slices are usable. We hope that our X-Diffusion (especially if published at ICLR 2025) will act as a first step towards this future we envision. We added more discussions in updated Appendix Section D.1 and D2.
>
> >_The related work section should cover super-resolution approaches. Also, the Full-Body MRI Analysis paragraph is very generic and pretty empty. See for instance Tunariu et al., British Journal of Radiology, 2020 on the use of whole-body MRI in clinical practice and Küstner et al., Radiology: Artificial Intelligence, 2020 for the automatic analysis methods related to these applications._
>
> We apologize for the minor issue in the related works section. We have modified the related works section to be more comprehensive and include the mentioned works and is now integrated in the paper PDF.
>
> >_Some references are too generic or do not correspond to the statement they are supposed to comfort, e.g. Tran et al., 2015; Sohl-Dickstein et al., 2015 or Kawar et al., 2022 in the second paragraph of the introduction._
>
> We apologize for the minor issues in the related works section and introduction references. They are all now fixed in the updated PDF .
>
> __Questions:__
> >_Please fix the hyperref links, many references in the Appendix do not point to the good figure, this makes the paper even more difficult to follow._
>
> All figures references are now fixed. Please see the updated PDF paper.

---

> > ### Author Response · Authors · 2024-11-25
> > **Follow-up**
> >
> > Dear reviewer,
> >
> > We would like to know if we adressed your comments. Please let us know whether we need to perform any additional experiments.

---

> > > ### Author Response · Authors · 2024-11-27
> > > **follow up**
> > >
> > > Dear reviewer fhnz,
> > >
> > > Today is the last day we can update the paper PDF. We would like to know if we addressed your comments. Please let us know whether we need to perform any additional experiments.
> > >
> > > Regards,

---

> > > > ### Comment · Reviewer_fhnz · 2024-11-27
> > > >
> > > > I would like to thank the authors for their thorough rebuttal. However I still see several limitations in the work presented.
> > > > - I still fail to see how such approach would be used in practice. Patients come for an imaging exam with a specific indication that triggers a specific acquisition protocol, so I do not see how the synthetic images generated would inform "the technician where to scan next" (and please avoid going so far as "potentially saving countless lives"). For example, predicting the next sequences to acquire to refine the diagnosis would match closer the objective targeted by the authors.
> > > > - The hallucinations are also a problem that would need to be further explored to assess the value of the presented work.
> > > >
> > > > Nonetheless, as the focus of ICLR is more on the methods than on the applications, and that the methods proposed are quite novel, I will raise my score from 3 to 5.

---

> ### Author Response · Authors · 2024-12-01
> **follow-up on feedback 1**
>
> >_I still fail to see how such approach would be used in practice. Patients come for an imaging exam with a specific indication that triggers a specific acquisition protocol, so I do not see how the synthetic images generated would inform "the technician where to scan next" (and please avoid going so far as "potentially saving countless lives"). For example, predicting the next sequences to acquire to refine the diagnosis would match closer the objective targeted by the authors._
>
> We evaluate this proposed protocol of active selection of slices on brain MRIs as follows. We use the first slice to reconstruct with X-Diffusion the full volume , then we use the tumour segmentation network to detect the brain tumour at the full 1-slice generated MRI, and the center of the tumour prediction determines the next slice we use , and repeat until the budget of slices is reached, these actively selected slices are then used with the X-Diffusion model trained to reconstruct the full MRI and report this results on test set compared to the naive random baseline reported in Table 1. The results are shown on the following table A (for PSNR reconstruction quality) and Table B (for Dice score of tumour compared to ground truth labels). It shows that such active selection simulating what can actually be performed in an MRI scanning session in the clinic creates better reconstruction quality for the active selection and obtains a better tumour detection compared to the naive randomly selected slices.
>
>
> ### Table A: PSNR Reconstruction Quality of using Active Selection strategy
>
> Number of Slices Used | Random Baseline PSNR | Active Selection PSNR
> -----------------------|----------------------|------------------------
> | 2                      | 27.21               | 28.79                 |
> | 5                      | 31.25               | 32.95                 |
> | 10                     | 33.27               | 33.84                 |
> | 31                     | 35.48               | 35.88                 |
>
>
>
> ### Table B: Dice Score for Tumour Detection of using Active Selection strategy
>
> | Number of Slices Used | Random Baseline Dice Score | Active Selection Dice Score |
> |-------------------------- |--------------------- |----------------------------- |
> | 2                        | 84.31              | 85.25                      |
> | 3                        | 84.16              | 85.04                      |
> | 5                        | 83.83              | 84.51                      |
> | 10                       | 82.87              | 82.96                      |
> | 31                       | 79.72              | 80.02                      |
>
>
> >_The hallucinations are also a problem that would need to be further explored to assess the value of the presented work._
>
> We addressed hallucinations briefly in Section 6.4 of the paper in which we showed when testing RATS-trained model (all BRATS samples have tumour) on the healthy brain MRIs of IXI dataset, the predicted tumor (using tumour segmentation network) is in 9.9% of the real healthy brains and in 11.3% of our generated brain MRIs.  This means the X-Diffusion model is mostly not hallucinating tumors when there is no tumor in the sample despite accurately reconstructing the healthy brains with a PSNR of f 35.86 .
>
> To test this hypothesis of dataset bias on the hallucination of X-Diffusion, we retrain X-Diffusion on BRATS but excluding all the slices that contain tumour (we have GT on BRATS) , and then test on the healthy brains of IXI. The resulting generated brains have a PSNR of 36.41 in reconstruction quality ( + 1.45 improvement in PSNR compared to training on the full dataset) , and a detection of tumour in 10.1 % of the generated samples in the test set ( -1.2% reduction in hallucinated tumour) . This gives an indication of why this hallucination occurs and how to mitigate it.
>
> >_Nonetheless, as the focus of ICLR is more on the methods than on the applications, and that the methods proposed are quite novel, I will raise my score from 3 to 5._
>
> We thank the reviewer for their considerations. We kindly ask the reviewer if all of their other concerns above were addressed properly and if they believe the paper is worthy of acceptance to reconsider raising the score to at least 6 which is still borderline acceptance but at least lowers the risk of rejecting the paper solely based on the clinical feasibility aspect of the paper (which is also addressed above).

---

### Official Review · Reviewer_F5BK · 2024-11-01

**Soundness:** 3
**Presentation:** 4
**Contribution:** 2
**Rating:** 6
**Confidence:** 4

**Summary:**

This paper introduces X-Diffusion, a novel MRI reconstruction algorithm that can generate 3D volumes by conditioning on single or few slice 2D MR input. In contrast to traditional MRI reconstruction, X-diffusion takes on a different challenge and tries to predict the entire 3D volume given the sparse (but uncorrupted/high-quality slice(s)). The authors conduct a variety of different experiments on multiple datasets and downstream tasks and test the meaningfulness of pathological findings in the reconstructed images. Given their experimental setting, they are able to challenge the state of the art.  In a variety of ablation studies, the authors assess the impact of individual key components, providing an explanation of the capability of the model.

**Strengths:**

Originality: While the proposed method is built upon recent advantages in computer vision, its combination of key components and area of application is novel. Especially the cross-sectional MRI synthesis approach, which enables the stacking of slices from arbitrary viewing directions, not just axial/coronal and sagittal planes, is of particular interest to the medical imaging community and constitutes - to the best of my knowledge - a unique contribution of the paper.

Quality: The paper is written in a very clear, concise, and comprehensive way. I believe the authors did a great job in setting their method into the context of related work, thoroughly and clearly describing it, and performing an interesting experimental section, including further experiments in the supplementary material. The ablations clarified and addressed questions that I previously had when reading the manuscript and addressed different important parameters of the pipeline, demonstrating the effectiveness of the key components, such as the stable diffusion encoder, etc.

Clarity: The presented method, related work, and experiments are on point. Ablations serve a purpose, the mathematical foundation behind key concepts was explained quite well, and figures and tables provided a great way of deepening the understanding of the reader. The supplementary offered a lot of great insights and the possibility for reproduction of experiments.

Significance: The method of synthesizing a 3D volume with a cross-sectional model from arbitrary viewing directions is very interesting to the medical imaging community. Also, X-Diffusion tackles a unique reconstruction scenario, which is very much unexplored.

**Weaknesses:**

1) Clinical relevance of the presented method:

While the clinicians in this paper argue that the model is clinically relevant - I have a conflicting opinion. While "scout" or "localizer" scans with a single or a few slices may be acquired and thus used for X-Diffusion, I believe it is dangerous to assume that these will substitute a real volumetric scan. While the authors argue that tumor information is preserved, I would rather say this information is correctly hallucinated. Especially small tumors and lesions in other neurodegenerative diseases, such as MS, often occur on a few or single slices. Given that single or few-slice acquisitions will not be able to capture this information- (at all) - I would argue only undersampled or lower-resolved images pose a clinical tradeoff between finding anomalies and reducing scan time. I feel deep learning-based reconstruction really introduces a well-motivated use case for super-resolution or reconstruction in these problem settings, especially when considering multiple (complementary contrasts), but - to me - single/few slices are far-fetched and thus X-Diffusion solves a rather unrealistic clinical setting? Consequently, I would dial this down in the manuscript and frame this more as an exploratory or proof of concept study, i.e., "confirmed the potential usefulness of the generated MRIs" feels a bit bold to me.

I would also be curious to see how the introduced method (key components would still have merit for these problem settings!) would perform on SR settings where we try to super-resolve anisotropic to isotropic scans. On another note, I acknowledge and really appreciate the study of k-space reconstruction. This poses a much more clinical use case, where X-diffusion is powerful but not on par (at least in scores) with SOTA (please correct me if I am wrong).

2) Difficulty in assessing how much is owed to conditioning with the "right" slices and the capability of the model:

While in regular medical image reconstruction, images are generally downsampled (in k-space) (c.f. ScoreMRI) or images with lower resolution are upscaled (TPDM), X-Diffusion has access to high-quality 2D slices that constitute a relevant part of the GT. In other words - the conditioning slices are not altered (i.e., they are identical and part of the GT). This is very different from the traditional scenarios - and thus poses a unique challenge in assessing the quality of the reconstruction. While the authors compare against SOTA baselines, these baselines were not originally designed to work as X-Diffusion works. This should be highlighted and discussed more prominently in the manuscript. I understand it is hard to come up with appropriate baselines, given that X-Diffusion is the first to attempt a single 2D slice to 3D reconstruction. However, the challenges that come with evaluating this should be more prominently discussed.

3) Downstream Evaluation:

Evaluation of the brain volume: While the brain volume estimation may serve as a first good downstream analysis task, I believe it constitutes an oversimplification of the problem. It would have been more meaningful to perform brain white-/grey matter analysis using, e.g., freesurfer on ground truth and reconstructed volume and comparing the results of this.

4) Please share the SSIM for all tables and experiments; these values are often more relevant to PSNR values for reconstruction.

5) A1 and A3 are almost identical. Consider merging or rewriting them to have one cohesive section.

**Questions:**

Clarification:

- I really appreciate the ablation for healthy patient inpainting, c.f. 6.4, but it would have been really important to know how many healthy slices were used for the inpainting experiment. How many tumors do we see if we only provide slices at the outer parts of the brain where tumors will not lie?

- For Figure 3, please clarify how many input slices are utilized from the target image. Considering the leftmost image is considered as input, am I right to assume that single-slice conditioning was utilized? However, given the crisp reconstruction, I would be surprised if this is the case. Was there tumor information in the conditioning - or is all of it impainted?

- For Figure 6, which image(s) were used to condition X-Diffusion - was any of the tumor information of the presented slice present in the slice? If so, it is not surprising that the X-diffusion model is able to recover this

- For Figure 9, Figure II/II Appendix (and more)  - how many slices were used to condition the model?

Suggestions:

- Please add SSIM scores alongside to the PSNR in the reconstruction experiments to the main paper (from supplementary).
- Consider adding deep learning-based metrics for image reconstruction, such as medical LPIPS (https://docs.monai.io/en/stable/losses.html#perceptualloss) or LPIPS (https://github.com/richzhang/PerceptualSimilarity), as these often correlate better with human perception and are widely used in medical image reconstruction as well.

---

> ### Author Response · Authors · 2024-11-21
> **Response to Reviewer's Comments and Questions**
>
> We thank the reviewer for their insightful comments and relevant questions, which we tried to address to the best of our ability. We updated the paper PDF with section D on Additional Discussions in supplementary material.
>
> >_1.Clinical relevance of the presented method:_
>
> We kindly agree with the reviewer that these generated “pseudo MRISs” are not a substitute for actual MRI scans as we described briefly in page 2 “It is important to note that the generated MRIs are not clinical replacements for true MRIs yet, but could provide a quick, affordable, and informative “pseudo-MRI" before conducting a full MRI examination.” We envision in the future this can be used in an active setup, while the patient is conducting a scan the model reconstructs the MRI and informs the technician where to scan next, reducing redundancy in the scanning process, and hence reducing the time in conducting the MRI scan, reducing the cost of running MRI scans, and potentially saving countless lives. We hope that our X-Diffusion (especially if published at ICLR 2025) will act as a first step towards this future we envision. We changed the tone a bit in the updated paper PDF to reflect your suggestion and added extended discussions about these points in Sections D.1 and D.2 of the updated appendix.
> While the experiments in Table I when the input slice is 31 are acting like a super-resolution problem with high-density input, and because our input aggregation layer is agnostic to the structure of the input, we indeed can perform SR from anisotropic to isotropic scans. We agree that the K-space setup is indeed a challenging case for our X-Diffusion, because the focus of the reconstruction from sparsity demands generalization and fewer details, while the K-space methods focus on high-frequency details.
>
> >_2.Difficulty in assessing how much is owed to conditioning with the "right" slices and the capability of the model:_
>
> We sincerely thank the reviewer for their insightful comment regarding the evaluation challenges of X-Diffusion compared to existing methods. We fully agree that conditioning on high-quality 2D slices that are part of the ground truth (GT) introduces unique considerations in assessing our model's capabilities versus the influence of the conditioning data. This point was briefly touched in the introduction of the paper and Fig1 of the main paper which highlights the difference in the setup.
> However,  to address this important point further, we have added a new subsection in D.3. In this subsection, we thoroughly discuss the evaluation challenges posed by the differences in input data and reconstruction objectives between X-Diffusion and traditional MRI reconstruction methods. We highlight how these differences impact the assessment of reconstruction quality and the comparative analysis with state-of-the-art (SOTA) baselines that were not originally designed to operate under the same conditions as X-Diffusion. We believe that this addition enhances the transparency of our work and provides a clearer understanding of how we have approached the evaluation of X-Diffusion, acknowledging the limitations and the steps taken to ensure a fair comparison.
>
> >_3. Downstream Evaluation:_
>
> We perform the requested analysis on white matter and we report the results in updated appendix sec D.14.
>
> >_4. Please share the SSIM for all tables and experiments; these values are often more relevant to PSNR values for reconstruction._
>
> We added SSIM and LPIPS in Tables that have only PSNR and included standard deviation along with PSNR values for main Table I and included it in Section D.7  in Tables X,XI,XII and XIII in the updated Appendix.
>
> >_5. A1 and A3 are almost identical. Consider merging or rewriting them to have one cohesive section._
>
> We fixed this in the updated PDF.
>
> >_How many tumors do we see if we only provide slices at the outer parts of the brain where tumors will not lie?_
>
> We run a similar study on BRATS in Table I in the Appendix.
>
> >_For Figure 3, please clarify how many input slices are utilized from the target image. Considering the leftmost image is considered as input, am I right to assume that single-slice conditioning was utilized?_
>
> Yes, it is from the input image slice on the left. The tumour coincided with some of the inputs but not all, Table I in the Appendix details.
>
> >_For Figure 6, which images were used to condition X-Diffusion_
>
> Because of space limitations, we are not able to include all the input images and how results and comparisons on the same figure. We show the input images now on a new figure XVI in appendix  Sec D.6
>
> >_For Figure 9, Figure II/II Appendix (and more) - how many slices were used to condition the model?_
>
> For Figure II/III of the appendix, we used our best X-Diffusion model conditioned on 31 slices at training to generate these volumes.

---

> > ### Comment · Reviewer_F5BK · 2024-11-25
> >
> > I would like to compliment the authors for writing a nice and clear rebuttal, addressing my questions, adding SSIM/LPIPS to the computation of the scores, and expanding further on some of the previously introduced ideas and experiments in the supplementary.
> >
> > Given that I see the clinical utility still somewhat limited (even though the authors transparently discuss it) I would like to keep my initial rating of the paper.

---

> > ### Author Response · Authors · 2024-11-25
> > **following response**
> >
> > __Suggestions:__
> > >_Please add SSIM scores alongside to the PSNR in the reconstruction experiments to the main paper (from supplementary)._
> >
> > Thank you for the suggestion, we have done it now, please see Section D.7.
> >
> > >_Consider adding deep learning-based metrics for image reconstruction, such as medical LPIPS or LPIPS_
> >
> > We add one Table X with LPIPS in the appendix Table in Sec D.8.

---

> ### Author Response · Authors · 2024-12-01
> **follow-up on feedback 1**
>
> >_Given that I see the clinical utility still somewhat limited (even though the authors transparently discuss it) I would like to keep my initial rating of the paper._
>
> We thank the reviewer for supporting this manuscript. While we acknowledged potential clinical future work that can be done to verify X-Doffusion feasibility on clinical setup on actual patients, we believe that would be suitable for a medical journal clinical study and is out of the scope of the current submission to ICLR conference which is about the method novelty, technical execution, and reconstructions results, where the reviewer agrees on (also reviewer `fhnz` just mentioned this observation). Hence we don’t see this clinical future work as a reason to lower the score of our paper which could result in a rejection given the current scores.
>
> However, we simulate such a clinical setup proposed protocol of active selection proposed by reviewer `fhnz`. We use the first slice to reconstruct with X-Diffusion the full volume , then we use the tumour segmentation network to detect the brain tumour at the full 1-slice generated MRI, and the center of the tumour prediction determines the next slice we use , and repeat until the budget of slices is reached, these actively selected slices are then used with the X-Diffusion model trained to reconstruct the full MRI and report this results on test set compared to the naive random baseline reported in Table 1. The results are shown on the following table A (for PSNR reconstruction quality) and Table B (for Dice score of tumour compared to ground truth labels). It shows that such active selection simulating what can actually be performed in an MRI scanning session in the clinic creates better reconstruction quality for the active selection and obtains a better tumour detection compared to the naive randomly selected slices.
>
> ### Table A: PSNR Reconstruction Quality of using Active Selection strategy
>
> | Number of Slices Used | Random Baseline PSNR | Active Selection PSNR |
> |------------------------|----------------------|------------------------|
> | 2                      | 27.21               | 28.79                 |
> | 5                      | 31.25               | 32.95                 |
> | 10                     | 33.27               | 33.84                 |
> | 31                     | 35.48               | 35.88                 |
>
>
>
> ### Table B: Dice Score for Tumour Detection of using Active Selection strategy
>
> | Number of Slices Used | Random Baseline Dice Score | Active Selection Dice Score |
> |--------------------------|---------------------|-----------------------------|
> | 2                        | 84.31              | 85.25                      |
> | 3                        | 84.16              | 85.04                      |
> | 5                        | 83.83              | 84.51                      |
> | 10                       | 82.87              | 82.96                      |
> | 31                       | 79.72              | 80.02                      |
>
> Based on the above, we kindly ask the reviewer if all of their other concerns were addressed properly and if they still believe the paper is worthy of acceptance to reconsider raising the score to 8 to lower the risk of rejecting the paper solely based on the clinical feasibility aspect of the paper.

---

> > ### Comment · Reviewer_F5BK · 2024-12-01
> >
> > Thank you once again for the very thorough rebuttal and the very thorough manuscript, I would really like too highlight and compliment you on the supplementary material.
> >
> > To me reviewing this paper has been challenging because it's not an easy decision. I like the rendering approach and the proposed method, but I am (still) critical regarding the utility of the approach. While there is always future work, and it's valid to put that up for discussion (or in the context of a journal extension), the initial paper should also provide a strong motivation why this approach seems interesting from an application point of view. While you transparently discuss the hallucination problem, I feel there is no straight forward mitigation of this problem. For cancer, where lesions are typically bigger, it's reasonable to assume a single slice or a view slices might be enough to project everything. For other pathological details, such as lesions in e.g. multiple sclerosis, where lesions might not span multiple slices, I believe this is a far stretch. Based on this reasoning, I have decided to keep my score.

---

### Official Review · Reviewer_Haqk · 2024-11-05

**Soundness:** 3
**Presentation:** 3
**Contribution:** 3
**Rating:** 6
**Confidence:** 3

**Summary:**

The authors introduces a new model for generating detailed 3D-MRI volumes from sparsified spatial-domain inputs, resulting in accurate 2D-to-3D scans reconstruction (in contrast to existing methods that require full 3D scans). The main idea is to incorporate view-dependent cross-sections during training, which is different from the current approaches.

**Strengths:**

The authors demonstrate superior performance in several benchmarking tasks, including brain tumor and full body MRIs.

The authors claim that the proposed model is able to generalize to novel domains, not seen during the training.

The problem, motivation and contributions are clearly stated.

Thorough experiments were conducted, methodically supporting the claims in paper. The idea of conditioning MRI reconstruction on different cross-sectional viewpoints seems novel.

**Weaknesses:**

There are no measurements on how fast the X-diffusion is. It would be beneficial to include comparisons with previous approaches. E.g., Figure 3 shows that for good quality reconstruction, almost T=1000 steps are required.

The limitations discussed in Sections 6 and 7 appear to be strong, potentially limiting the practical value of the work.

The authors dismiss the study of whether it is possible to reduce the number T in diffusion without sacrificing the quality? And how would this reduction be specific to medical imaging domain? (can simply substitute ideas of speeding up from text-to-image models?)

**Questions:**

It would be interesting to see how X-Diffusion works with other types of augmentations, like blurring, changing colorspace etc. Intuitively, this should boost ability to generalize OOD or at least improve algorithm compared to using only SO3.

Author’s do not discuss what are the benefits of using pretrained VAE from SD without any additional finetuning (i.e VAE is frozen) to specific MRI domain. Additional discussion would be great.

Is it possible to integrate and check performance using existing text-to-3D models. Seems like utilizing directly 3D generative model provides more inductive bias, which should improve learning.

---

> ### Author Response · Authors · 2024-11-21
> **Response to Reviewer's Comments and Questions**
>
> We thank the reviewer for their insightful comments and relevant questions, which we tried to address to the best of our ability. We updated the paper PDF with section D on Additional Discussions for the rebuttal.
>
> >_There are no measurements on how fast the X-diffusion is. It would be beneficial to include comparisons with previous approaches. E.g., Figure 3 shows that for good quality reconstruction, almost T=1000 steps are required._
>
> Table VI in the original appendix page 26 shows compute cost and runtimes that are measured on a computer with a single GPU a6000, 48GB of RAM. The full sample would require slightly over 2 minutes. The 1000 steps are standard followed on LDM, Stable Diffusion, etc. Section D.11 and Figure XVII in the updated appendix include additional analysis of the time steps T, indicating we do not need 1000 steps to actually converge to good solutions.
>
> >_The limitations discussed in Sections 6 and 7 appear to be strong, potentially limiting the practical value of the work._
>
> While we agree that the diffusion model can hallucinate in some cases to account for missing information, we conduct a conformal prediction study to assess the confidence of the predictions that can be used by doctors and practitioners to assess the confidence in the diffusion output. We show results in the updated Appendix section D.13.
>
> >_The authors dismiss the study of whether it is possible to reduce the number of T in diffusion without sacrificing the quality? And how would this reduction be specific to the medical imaging domain? (can simply substitute ideas of speeding up from text-to-image models?)_
>
> We conduct an ablation study on the number of time steps in Section D.12 and Figure D.12 in the updated appendix, which includes additional analysis. We also try to use the recent diffusion models like SD 2.1 and SD XL to speed up the diffusion process or increase performance and report results in Section D.10 and Table 12 of the updated Appendix.
>
>
> __Questions:__
>
> >_It would be interesting to see how X-Diffusion works with other types of augmentations, like blurring, changing colorspace etc. Intuitively, this should boost the ability to generalize OOD or at least improve the algorithm compared to using only SO3._
>
> The SO(3) transformation is not only used as an augmentation but also generalizes the learning of volumes because it exposes the diffusion model to internal structure from different views and we aggregate the final prediction from all different directions. We try different augmentations: rotation, flip, and scale to improve the diffusion output at test time and report the results in Sec D.9 and Table XI.
>
> >_Author’s do not discuss what are the benefits of using pretrained VAE from SD without any additional finetuning (i.e VAE is frozen) to specific MRI domain. Additional discussion would be great._
>
> While we indeed use VAE from SD, it is trained to reconstruct any image from the latent space with high accuracy. More modern approaches have been proposed like in SD2.1 and SD XL. We added this in the discussion and X-Diffusion results on these modern approaches in Sec D.10 and Table 12 in the updated appendix.
>
> >_Is it possible to integrate and check performance using existing text-to-3D models? Seems like utilizing directly 3D generative model provides more inductive bias, which should improve learning._
>
> The use of modern Text-to-3D or image-to-3D that is based on Video Diffusion models or Multi-view diffusion models (eg.g  SDV3D[a])  to reconstruct the MRI in the X-Diffusion pipeline is indeed a very promising direction to improve the performance. We expect this to work slightly better than the current X-Diffusion despite struggling with computing and memory cost issues, and we leave the dedicated investigation of this direction for future work.
> [a] Sv3d: Novel multi-view synthesis and 3d generation from a single image using latent video diffusion, (ECCVC 2024).

---

> > ### Author Response · Authors · 2024-11-25
> > **Follow-up**
> >
> > Dear reviewer,
> >
> > We would like to know if we adressed your comments. Please let us know whether we need to perform any additional experiments.

---

> > > ### Comment · Reviewer_Haqk · 2024-11-26
> > >
> > > Thank you for the rebuttal. I believe the new experiments mitigate my critiques but the limitations are still strong for practical implementation of the method. Reviewer fhnz writes along these lines too w.r.t. clinical validation (if only a bit too relentlessly to my taste). I thus keep my original score of the borderline accept.

---

> ### Author Response · Authors · 2024-12-01
> **response on feedback 1**
>
> >_Thank you for the rebuttal. I believe the new experiments mitigate my critiques but the limitations are still strong for practical implementation of the method. Reviewer fhnz writes along these lines too w.r.t. clinical validation (if only a bit too relentlessly to my taste). I thus keep my original score of the borderline accept._
>
> We thank the reviewer for supporting this manuscript. While we acknowledged potential clinical future work that can be done to verify X-Doffusion feasibility on clinical setup on actual patients, we believe that would be suitable for a medical journal clinical study and is out of the scope of the current submission to ICLR conference which is about the method novelty, technical execution, and reconstructions results, where the reviewer agrees on (also reviewer `fhnz` just mentioned this observation). Hence we don’t see this clinical future work as a reason to lower the score of our paper which could result in a rejection given the current scores.
>
> However, we simulate such a clinical setup proposed protocol of active selection proposed by reviewer `fhnz`. We use the first slice to reconstruct with X-Diffusion the full volume , then we use the tumour segmentation network to detect the brain tumour at the full 1-slice generated MRI, and the center of the tumour prediction determines the next slice we use , and repeat until the budget of slices is reached, these actively selected slices are then used with the X-Diffusion model trained to reconstruct the full MRI and report this results on test set compared to the naive random baseline reported in Table 1. The results are shown on the following table A (for PSNR reconstruction quality) and Table B (for Dice score of tumour compared to ground truth labels). It shows that such active selection simulating what can actually be performed in an MRI scanning session in the clinic creates better reconstruction quality for the active selection and obtains a better tumour detection compared to the naive randomly selected slices.
>
> ### Table A: PSNR Reconstruction Quality of using Active Selection strategy
>
> | Number of Slices Used | Random Baseline PSNR | Active Selection PSNR |
> |------------------------|----------------------|------------------------|
> | 2                      | 27.21               | 28.79                 |
> | 5                      | 31.25               | 32.95                 |
> | 10                     | 33.27               | 33.84                 |
> | 31                     | 35.48               | 35.88                 |
>
>
>
> ### Table B: Dice Score for Tumour Detection of using Active Selection strategy
>
> | Number of Slices Used | Random Baseline Dice Score | Active Selection Dice Score |
> |--------------------------|---------------------|-----------------------------|
> | 2                        | 84.31              | 85.25                      |
> | 3                        | 84.16              | 85.04                      |
> | 5                        | 83.83              | 84.51                      |
> | 10                       | 82.87              | 82.96                      |
> | 31                       | 79.72              | 80.02                      |
>
> Based on the above, we kindly ask the reviewer if all of their other concerns were addressed properly and if they still believe the paper is worthy of acceptance to reconsider raising the score to 8 to lower the risk of rejecting the paper solely based on the clinical feasibility aspect of the paper.

---

### Author Response · Authors · 2024-11-25
**Added section D in the appendix dedicated for the rebuttal**

The new subsection D in the appendix is dedicated to the rebuttal requested experiments. visualizations and additional discussions that came up during the reviews. It was originally only included in the updated supplementary materials (on Nov 21) but is now also included in the main updated PDF paper as part of the appendix Sec D (on Nov 25). We hope that did not cause any confusion or missing the new subsection D completely.

It is also important to note that this work is a paradigm shift in MRI reconstruction, a new way of reconstructing and generating medical images from sparse inputs with AI. Hence, it will get a lot of resistance from the traditional school of medical imaging and from physicians, similar to how MRI originally received so much resistance for adoptions in the 1980s. We hope the reviewers and Area chairs take this into account when evaluating our work and see how many potential human lives will be saved in the future by fostering this line of work.

---

> ### Author Response · Authors · 2024-12-01
> **final general comments**
>
> Based on the discussions below, all of the reviewers initial concerns and questions are cleared during rebuttal except for the clinical aspect of the paper. While we acknowledged potential clinical future work that can be done to verify X-Doffusion feasibility on clinical setup on actual patients, we believe that would be suitable for a medical journal clinical study and is out of the scope of the current submission to ICLR conference which is about the method novelty, technical execution, and reconstructions results, where the reviewers agree on (also reviewer `fhnz` just mentioned this observation). Hence we don’t see this clinical future work as a reason to lower the score of our paper.
>
> However, we simulate such a clinical setup proposed protocol of active selection proposed by reviewer `fhnz`. We use the first slice to reconstruct with X-Diffusion the full volume , then we use the tumour segmentation network to detect the brain tumour at the full 1-slice generated MRI, and the center of the tumour prediction determines the next slice we use , and repeat until the budget of slices is reached, these actively selected slices are then used with the X-Diffusion model trained to reconstruct the full MRI and report this results on test set compared to the naive random baseline reported in Table 1. The results are shown on the following table A (for PSNR reconstruction quality) and Table B (for Dice score of tumour compared to ground truth labels). It shows that such active selection simulating what can actually be performed in an MRI scanning session in the clinic creates better reconstruction quality for the active selection and obtains a better tumour detection compared to the naive randomly selected slices.
>
> ### Table A: PSNR Reconstruction Quality of using Active Selection strategy
>
> | Number of Slices Used | Random Baseline PSNR | Active Selection PSNR |
> |------------------------|----------------------|------------------------|
> | 2                      | 27.21               | 28.79                 |
> | 5                      | 31.25               | 32.95                 |
> | 10                     | 33.27               | 33.84                 |
> | 31                     | 35.48               | 35.88                 |
>
>
>
> ### Table B: Dice Score for Tumour Detection of using Active Selection strategy
>
> | Number of Slices Used | Random Baseline Dice Score | Active Selection Dice Score |
> |--------------------------|---------------------|-----------------------------|
> | 2                        | 84.31              | 85.25                      |
> | 3                        | 84.16              | 85.04                      |
> | 5                        | 83.83              | 84.51                      |
> | 10                       | 82.87              | 82.96                      |
> | 31                       | 79.72              | 80.02                      |

---

### Meta-Review · Area_Chair_gT9j · 2024-12-25

**Metareview:**

This manuscript proposes diffusion model based volumetric conditional generation (inpainting with extremely sparse inputs) intended for MRI (structural contrasts, presumably). They provide results on a variety of datasets, including UK Biobank (whole body MR), BRATS (brain tumor public challenge data), IXI (healthy human brain data), and the fastMRI knee data.

Overall, reviewer discussion is focused on experiments and clinical viability. There are only a few notes from reviewers about the method itself; it appears to be exactly conditional latent diffusion with an augmented training process. Reviewer opinion is mixed on viability; while reported results have high PSNR which is quite promising, several reviewers voice concerns that PSNR does not inform well upon important features (and their possible changes). Both reviewer `YwSq` and `F5BK` state as much, with `F5BK` suggesting LPIPS or other metrics, and `YwSq` highlighting hallucinated tissue/ventricle/tumor differences between volumes with different inputs. These are addressed in an additional Appendix (D, and subsections therein).

Further, reviewers also find the broader use case to be somewhat unrealistic. (paraphrasing: "when would you go in for only a partial scan?").

In summary, results appear quite astonishing in the applied domain, though metrics may hide large issues, the methodology itself is unremarkable given prior work, and the broader context is questionable by some. By scores, this is definitionally borderline for ICLR. I do not think ICLR is the correct venue to discuss the applied results as is necessary for understanding the authors' claim that "this work is a paradigm shift in MRI reconstruction"; the conference in general does not have the correct audience nor (relatedly) the depth of domain knowledge to make that assessment even if the paper is published in the proceedings. Without major methodological contribution there is less relevant contribution outside of those domain specific advancements, and thus I am choosing to recommend to reject this paper. I will defer to the SACs/PC Chair's final decision in either case.

**Additional Comments On Reviewer Discussion:**

I would like to add that PSNR in a [0,1] normalization scheme is a very bad metric for MRI in particular. Viewing the intensity histogram of a typical T1w MRI will show extreme outliers in intensity for most scanner vendor/system combinations. A few voxels will have extreme intensity due to the acquisition scheme (k-space) and RF coil dynamic range (which has much larger bit-depth than processed natural images). Without more nuanced normalization, re-normalization of the histogram by dividing by max intensity will likely produce a compressed effective dynamic range while keeping the reported range at [0,1], leading to increased PSNR for degenerate solutions. This is exacerbated in contrast enhanced scans, such as those found in BRATS.

On the otherhand, I commend the authors for providing many alternative metrics. While these sometimes suffer from similar problems, they at least have facilitated discussion around their reported experimental error.

---

### Decision · Program_Chairs · 2025-01-22

Reject